# Endocannabinoid signalling modulates susceptibility to traumatic stress exposure

Rebecca J. Bluett[1,2,*], Rita Báldi[1,*], Andre Haymer[1], Andrew D. Gaulden[1], Nolan D. Hartley[1,2], Walker P. Parrish[3], Jordan Baechle[1], David J. Marcus[1,2], Ramzi Mardam-Bey[1], Brian C. Shonesy[3], Md. Jashim Uddin[4], Lawrence J. Marnett[4,5], Ken Mackie[6], Roger J. Colbran[2,3,7,8], Danny G. Winder[1,2,3,7,8] & Sachin Patel[1,2,3,7,8]

Stress is a ubiquitous risk factor for the exacerbation and development of affective disorders including major depression and posttraumatic stress disorder. Understanding the neurobiological mechanisms conferring resilience to the adverse consequences of stress could have broad implications for the treatment and prevention of mood and anxiety disorders. We utilize laboratory mice and their innate inter-individual differences in stress-susceptibility to demonstrate a critical role for the endogenous cannabinoid 2-arachidonoylglycerol (2-AG) in stress-resilience. Specifically, systemic 2-AG augmentation is associated with a stress-resilient phenotype and enhances resilience in previously susceptible mice, while systemic 2-AG depletion or CB1 receptor blockade increases susceptibility in previously resilient mice. Moreover, stress-resilience is associated with increased phasic 2-AG-mediated synaptic suppression at ventral hippocampal-amygdala glutamatergic synapses and amygdala-specific 2-AG depletion impairs successful adaptation to repeated stress. These data indicate amygdala 2-AG signalling mechanisms promote resilience to adverse effects of acute traumatic stress and facilitate adaptation to repeated stress exposure.

[1] Department of Psychiatry and Behavioral Sciences, Vanderbilt University Medical Center, Nashville, Tennessee 37232, USA. [2] The Vanderbilt Brain Institute, Vanderbilt University Medical Center, Nashville, Tennessee 37232, USA. [3] Department of Molecular Physiology & Biophysics, Vanderbilt University Medical Center, Nashville, Tennessee 37232, USA. [4] A.B. Hancock Jr. Memorial Laboratory for Cancer Research and Vanderbilt Institute of Chemical Biology, Nashville, Tennessee 37232, USA. [5] Departments of Biochemistry, Chemistry and Pharmacology, Vanderbilt University Medical Center, Nashville, Tennessee 37232, USA. [6] Department of Psychological and Brain Sciences, Indiana University, Bloomington, Indiana 47405, USA. [7] Vanderbilt Kennedy Center for Human Development, Nashville, Tennessee 37232, USA. [8] Vanderbilt Center for Addiction Research, Vanderbilt University, Nashville, Tennessee 37232, USA. * These authors contributed equally to this work. Correspondence and requests for materials should be addressed to S.P. (email: sachin.patel@vanderbilt.edu).

Stress is a major risk factor for neuropsychiatric disease, particularly major depression and anxiety disorders, and is etiologically causal in posttraumatic stress disorder (PTSD)[1–7]. Stress-resilience is associated with reduced risk of psychopathology and is an active process of adaptation, not merely the absence of maladaptive changes induced by stress exposure[8–13]. Understanding the biological mechanisms promoting stress-resilience could lead to novel treatments for stress-related psychiatric disorders. Here we elucidate a role for endogenous cannabinoid (eCB) 2-arachidonoylglycerol (2-AG) in promoting resilience to acute traumatic stress and successful adaptation to repeated homotypic stress exposure.

The eCB system is composed of the presynaptic cannabinoid CB1 receptor (CB1R), its endogenous ligands including anandamide (arachidonoylethanolamine; AEA) and 2-AG, and enzymes mediating eCB turnover[14,15]. Neuronal 2-AG is synthesized postsynaptically primarily by diacylglycerol lipase α (DAGLα)[16,17], while AEA can be generated via multiple enzymatic cascades[18]. After release from the postsynaptic compartment, eCBs travel retrogradely to the presynaptic terminal where they bind CB1Rs, which when activated reduce vesicular neurotransmitter release from the synaptic terminal[15,19]. 2-AG is primarily degraded presynaptically by monoacylglycerol lipase (MAGL), while AEA is degraded postsynaptically by fatty acid amide hydrolase (FAAH)[15,18], and pharmacological inhibition of MAGL or FAAH can increase 2-AG or AEA-mediated eCB signalling, respectively.

eCBs have been implicated in modulating anxiety, fear learning and stress responsivity[20–22]. Pharmacological augmentation of AEA signalling reduces unconditioned anxiety and reduces stress-induced increases in anxiety-like behaviour, corticosterone release, and dendritic remodelling[20]. AEA augmentation also facilitates extinction learning in mice[20]. Furthermore, stress exposure can decrease brain AEA levels, which are inversely correlated with severity of stress-induced anxiety-like behaviours[23]. Although compelling evidence suggests that AEA signalling buffers against stress-related affective pathology[20,24], the role of 2-AG signalling in stress-modulation is only now becoming appreciated. For example, pharmacological augmentation of 2-AG signalling can reduce unconditioned anxiety and prevent emergence of stress-induced anxiety-like behaviours[25–29], while genetic 2-AG deficiency results in increased anxiety-like behaviours[16,30]. Moreover, chronic homotypic stressors increase 2-AG levels within the amygdala and prefrontal cortex[31,32]. Despite these findings, whether 2-AG signalling within these regions regulates resilience to traumatic stress exposure has not been investigated. To directly address this critical question, herein we develop and validate a model for rapid evaluation of inter-individual differences in stress-resilience. We then utilize pharmacological and circuit-specific electrophysiological approaches combined with a novel conditional genetic model to demonstrate a key role for 2-AG signalling in promoting stress-resilience and successful adaptation to repeated stress exposure.

## Results

**Augmenting 2-AG reduces stress-induced anxiety-like behaviour.** To begin to elucidate the role of 2-AG signalling in modulating stress-resilience, we first determined the effects of systemic pharmacological 2-AG augmentation on stress-induced anxiety-like behaviours using the novelty-induced hypophagia (NIH) test, which is highly sensitive to acute traumatic stress and eCB manipulation[23,33]. Acute administration of the MAGL inhibitor JZL-184 (8 mg kg$^{-1}$) increased 2-AG and decreased its metabolite, arachidonic acid (AA), without significantly affecting AEA in several limbic brain regions (Fig. 1a–c). JZL-184 significantly reduced anxiety-like behaviour two hours after administration, measured as a reduction in latency to consume palatable food in the NIH test 24 h after one or five days of foot-shock stress, but not in unstressed mice (Fig. 1d). JZL-184 also increased consumption following one day of stress (Fig. 1e). The CB1R inverse agonist Rimonabant blocked the effects of JZL-184 after five days of stress (Fig. 1d,e, diagonal stripes). Visual inspection of the cumulative distribution curves of feeding latency for vehicle vs. JZL-184 revealed larger separation at higher latencies (Fig. 1f–h), suggesting JZL-184 preferentially reduced the number of mice exhibiting high feeding latencies after stress. Rimonabant alone significantly increased latency and reduced consumption after one or five days of stress (Fig. 1i–k). Taken together, these data suggest bidirectional modulation of stress-induced anxiety states by enhancing versus inhibiting 2-AG-CB1R signalling.

Given the well-known effects of eCB signalling on food intake[34], we confirmed the anxiolytic effects of JZL-184 after stress exposure in another validated assay independent of appetitive motivation. Specifically, JZL-184 significantly increased light-time, light-distance, and % light-distance, and decreased latency to enter the light-zone, without significantly altering dark-distance in the light-dark test 24 h after acute stress (Fig. 1l). Furthermore, JZL-184 treatment did not affect food consumption 24 h after acute stress under less aversive (low light) NIH testing conditions, or increase weight gain over 10 days of treatment (Supplementary Fig. 1). JZL-184 also did not significantly impact locomotor behaviour in an open-field test at baseline or after five days of stress (Supplementary Fig. 1). These data indicate that the ability of JZL-184 to reduce stress-induced anxiety-like behaviour in the NIH test is not related to enhanced appetitive or consummatory drive *per se*, or a result of altered locomotor activity.

**2-AG augmentation promotes a stress-resilient phenotype.** Our data thus far indicate that 2-AG-CB1R signalling modulates stress-induced anxiety-like behaviour; however, whether 2-AG signalling affects susceptibility to the development of stress-induced anxiety-like behaviour is still unclear. To explicitly test this hypothesis, we developed a repeated NIH testing paradigm, which allowed for the detection of inter-individual differences in stress susceptibility (Fig. 2a). Six days after baseline novel-cage testing, mice were foot-shock stressed and, 24 h later, evaluated in a 2nd novel-cage test. The change in each individual's latency between baseline and post-stress testing is shown in the population distribution in Fig. 2b. Examination of the population distribution of stress-induced changes in latency revealed a bimodal distribution, with data significantly better fit to two independent Gaussian distributions (resilient $n = 77$, susceptible $n = 43$, $F_{(3,144)} = 112.3$, $P < 0.0001$ Extra Sum-of-squares F test, Fig. 2c). Taking into consideration the observed anti-node between the two distributions in Fig. 2b, our aim to generate a meaningful susceptible group (i.e., not affected by low responders), and previous studies indicating that a difference in feeding latency on the order of 1–3 min represents a biologically relevant difference in anxiety-like behaviour[35], we empirically divided the population into susceptible and resilient groups. Susceptible mice were defined as having a stress-induced change in latency $\geq 120$ s, while those with a change in latency $< 120$ s were categorized as resilient. The means of the major and minor distributions after this categorization were $-21$ and $168$ s, respectively. Figure 2c,d illustrate this cutoff, splitting individuals into stress resilient and susceptible populations. Retrospective examination of baseline NIH latencies of the two groups revealed that their distributions overlapped considerably

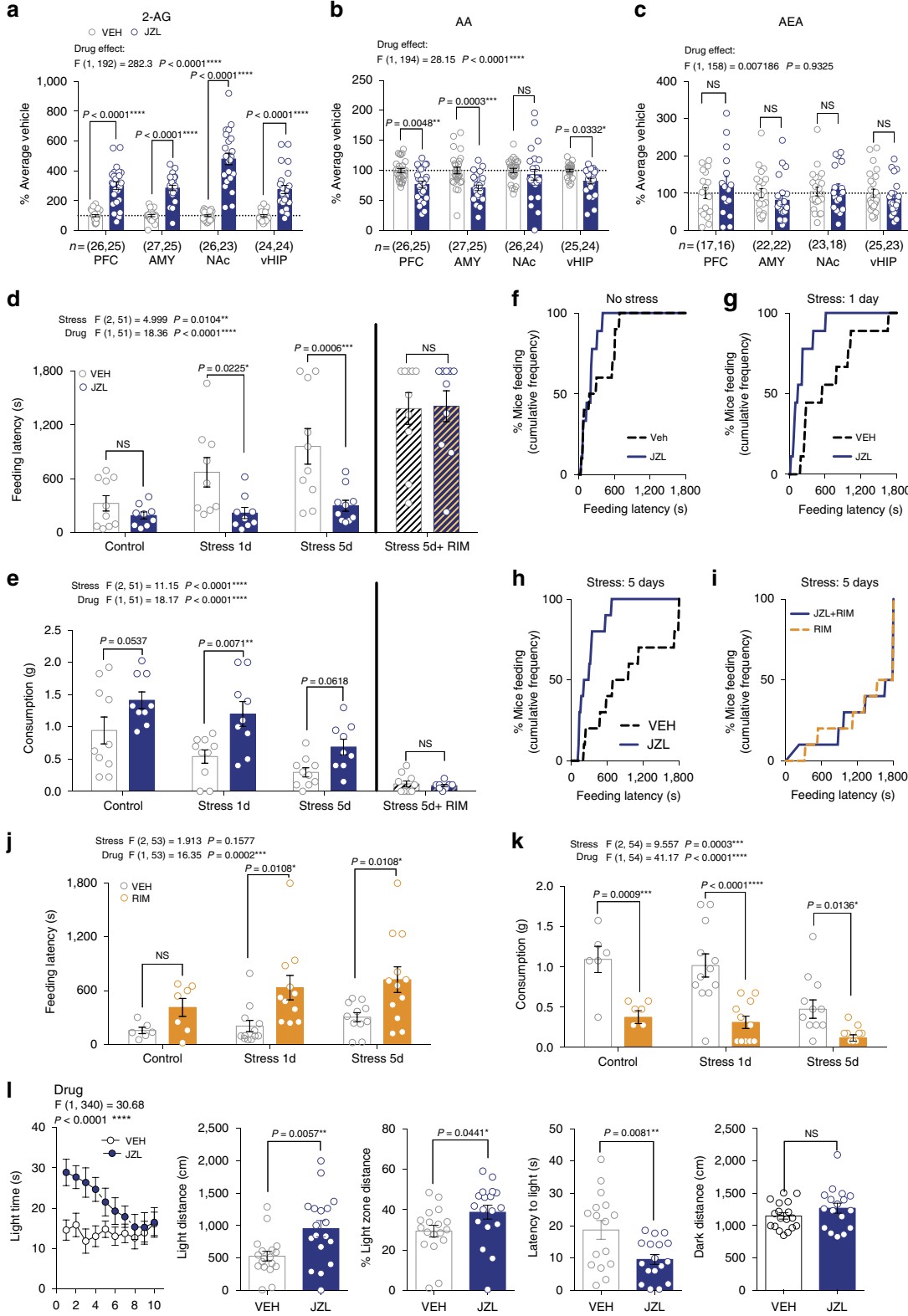

**Figure 1 | Modulation of stress-induced anxiety-like behaviour by 2-AG signalling.** (**a**–**c**) Effects of JZL-184 (8 mg kg$^{-1}$; blue) on 2-arachidonoylglycerol (2-AG), arachidonic acid (AA), and anandamide (AEA) in the prefrontal cortex (PFC), amygdala (AMY), nucleus accumbens (NAc), and ventral hippocampus (vHIP). Data combined from two independent experiments. (**d**,**e**) Effects of JZL-184 treatment on feeding latency (top) and consumption (bottom) in the novelty-induced hypophagia test (NIH) without stress, after 1 or 5 days of foot-shock stress, and after 5 days of stress in combination with the CB1R inverse agonist Rimonabant (RIM; 1 mg kg$^{-1}$). (**f**–**i**) Cumulative feeding latency distributions of vehicle and JZL-184-treated mice without stress, after 1 or 5 days of foot-shock stress, and after 5 days of stress in combination with Rimonabant. (**j**,**k**) Effects of Rimonabant (orange) on feeding latency and consumption in NIH without stress, and after 1 and 5 days of foot-shock stress. (**i**) Effects of JZL-184 treatment in the light–dark box test after 1 day of foot-shock stress. F and P values for two-way ANOVA shown above (**a**–**e**,**j**–**l**). P values shown for pairwise comparisons derived from Holm-Sidak multiple comparisons test after ANOVA or unpaired two-tailed t-test (**l**). Data are presented as mean ± s.e.m.

(resilient mean 258 s, susceptible mean 203 s), indicating that baseline anxiety-like behaviour does not predict stress susceptibility (Fig. 2e,f). Interestingly, specifically for the resilient subpopulation, there was a significant correlation

between baseline latency and post-stress reduction in latency, such that the mice exhibiting the highest baseline latencies showed the largest decrease after stress, confirming the resilient nature of this group (Fig. 2g). There was no correlation between

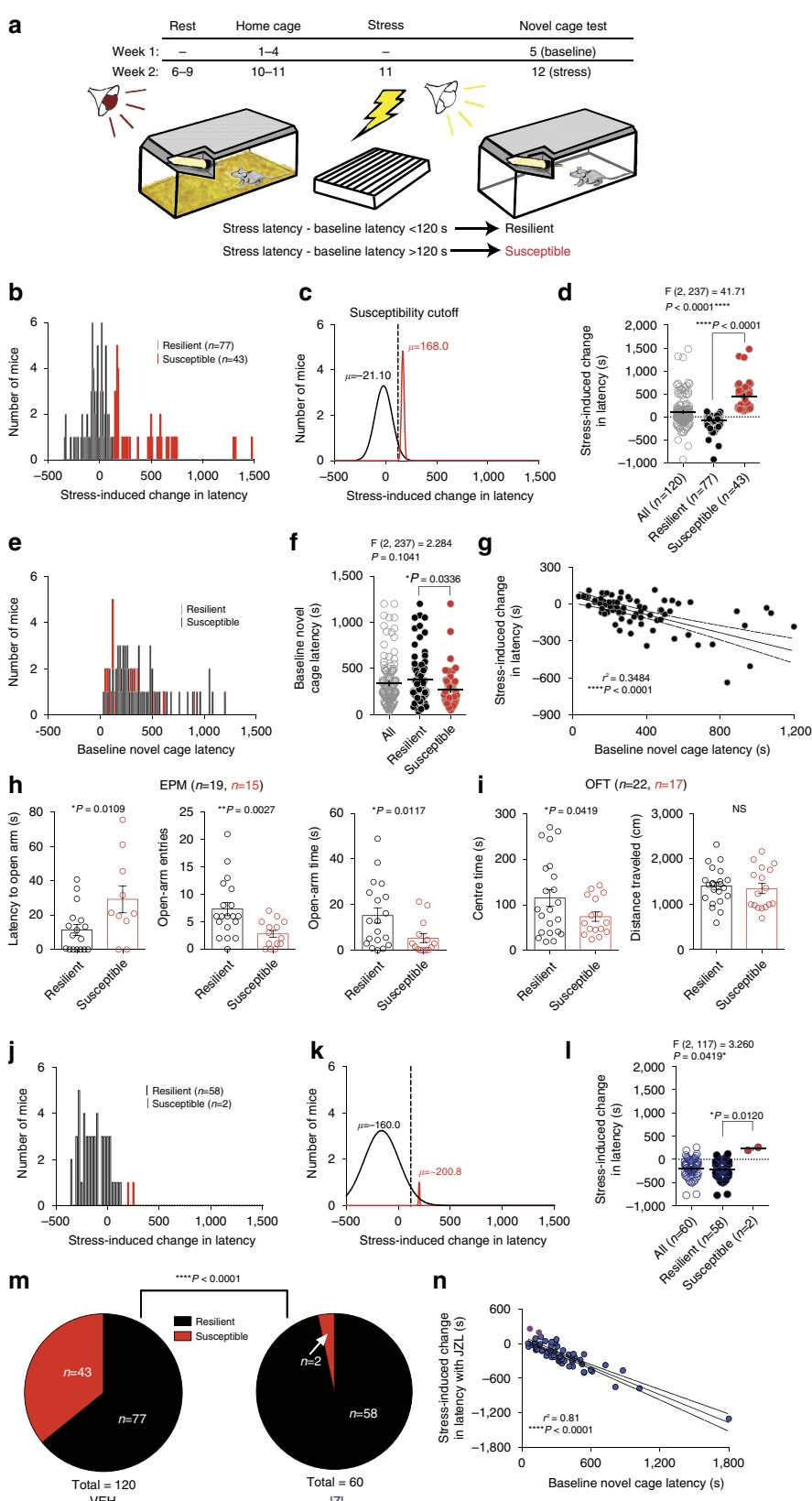

baseline latency and stress-induced increase in latency for susceptible mice ($n = 43$, $r^2 = 0.0097$, $P = 0.529$, linear regression).

To further validate phenotypic separation based on stress-induced NIH latency changes, one cohort of mice was categorized as resilient or susceptible and tested 7 days later in the elevated plus maze (EPM) and open-field test (OFT) 24 h after a 2nd stress exposure. As expected, susceptible mice exhibited higher stress-induced anxiety-like behaviour than resilient mice in the EPM and OFT (Fig. 2h,i). In a separate cohort, EPM and OFT were conducted before stress, and mice classified *post hoc* as susceptible or resilient using the procedure described above. Using this approach, we show conclusively that baseline (pre-stress) anxiety-like behaviour does not significantly differ between groups before stress exposure (Supplementary Fig. 2). Additionally, to be certain that our results were not due to differences in sensory processing of the foot-shock itself we measured foot-shock sensitivity thresholds and behavioural responses to foot-shock stress exposure and found no differences between resilient and susceptible groups (Supplementary Fig. 2). Fear learning and recall also did not differ between groups, as indicated by percent freezing across two foot-shock sessions with week one tone 6 (T6) freezing indicative of within-session learning, week two baseline (BL) freezing indicative of between-session context recall, and week two tone 1 (T1) freezing indicative of context + tone recall (Supplementary Fig. 2). To determine if susceptibility was primarily due to differential HPA axis responsivity, we also measured stress-induced corticosterone immediately following the week two foot-shock stress session and found no group differences or correlation between stress-induced corticosterone and either NIH susceptibility or foot-shock responsivity (Supplementary Fig. 2). These data support the segregation of two populations of mice without differences in basal anxiety-like behaviour or acute shock-responsivity, but with differential sensitized anxiety-like behavioural sequelae of stress exposure.

We next wanted to determine if systemic 2-AG augmentation could promote stress-resilience in this model. To examine this experimentally, baseline feeding latencies were obtained, followed seven days later by 24 h-post-stress novel-cage testing with JZL-184 treatment. JZL-184 treatment before the stress-test shifted the distribution of stress-induced changes in latency toward resilience, nearly eliminating the susceptible subpopulation (Fig. 2j–m). Analysis of the overall distribution of stress-induced changes in latency between JZL-184 and vehicle treatment revealed a dramatic increase in the resilient proportion at the expense of the susceptible population (Fig. 2b,c versus j,k, and see ratios in

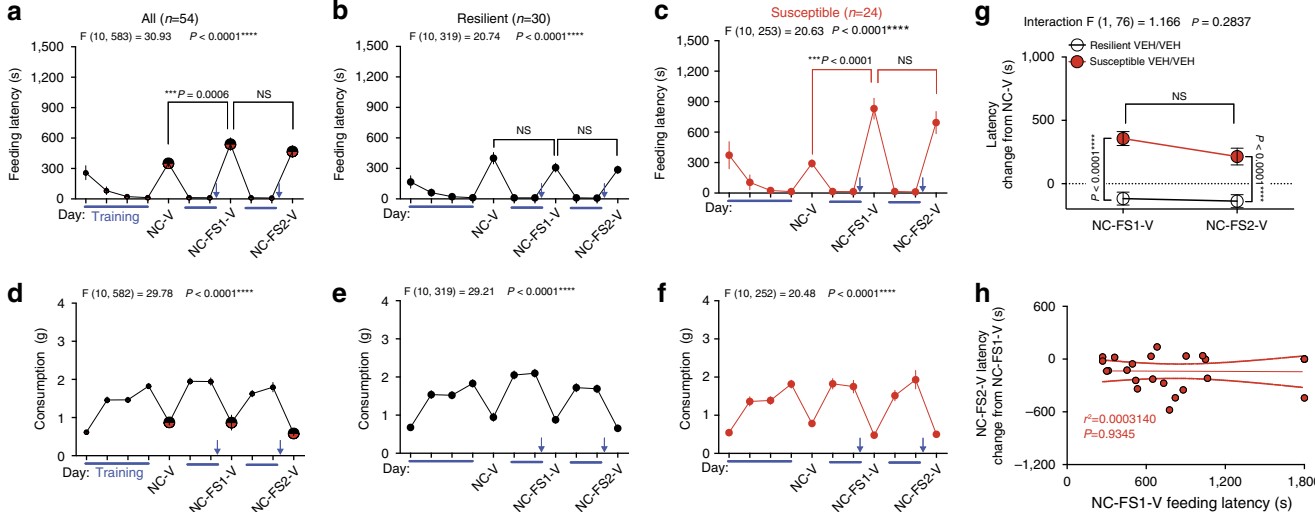

**Figure 3 | Stress susceptibility is a stable trait.** (**a**–**c**) Home cage training (blue lines) and novel cage (NC-V) latencies and (**d**–**f**) consumption across one baseline and two novel-cage foot-shock stress tests (NC-FS1-V and NC-FS2-V) with vehicle treatment. (**g**) Direct comparison of the latency change from baseline between the two post-stress NIH novel cage tests. (**h**) Correlation between the first stress-test latency and the change in latency between the 2nd and 1st stress novel-cage tests for susceptible individuals. Blue arrows indicate foot-shock stress exposure. F and P values for one-way (**a**–**f**) or two-way (**g**) ANOVA shown above individual panels. P values shown for pairwise comparisons from Holm-Sidak multiple comparisons test after ANOVA. $R^2$ and P value for linear regression shown in **h**. Data are presented as mean ± s.e.m.

**Figure 2 | Elevating 2-AG shifts the distribution of stress-susceptibility toward resilience.** (**a**) Schematic of behavioural paradigm. (**b**) Histogram of stress-induced change in latency (stress latency minus baseline latency) to consume in the NIH novel-cage test. (**c**) Gaussian curves fitting the resilient (black) and susceptible (red) subpopulations. Dashed line indicates 120-second post-stress latency increase susceptibility cutoff. (**d**) Stress-induced change in latency in the whole population and split into susceptible and resilient subgroups. (**e**) Histogram of pre-stress novel cage latencies categorized by resilience. (**f**) Individuals' pre-stress novel-cage latencies. (**g**) Correlation of resilient subpopulation's baseline and post-stress changes in latency. (**h**) Elevated plus maze (EPM) and (**i**) open-field test (OFT) measured 24 h after foot-shock stress, one week after susceptibility characterization. (**j**) Histogram of 24 h post-stress changes in latency with JZL-184 treatment 2 h before testing. (**k**) Gaussian distributions for resilient and susceptible subpopulations with JZL-184 treatment. (**l**) Stress-induced change in latency in the whole population and split into susceptible and resilient subgroups with JZL-184 treatment. (**m**) Proportion of susceptible and resilient mice after either vehicle (VEH) or JZL-184 treatment. (**n**) Correlation between pre-stress latencies and stress-induced changes in latency with JZL-184. Data in **a**–**g** was aggregated from 3 cohorts of 40 mice that were used for subsequent experiments (see Methods for details). F and P values for one-way ANOVA shown above (**d**,**f**,**l**). P values shown for pairwise comparisons derived from Sidak multiple comparisons test after ANOVA (**d**,**f**,**l**) or unpaired one-tailed *t*-test (**h**,**i**) shown in each panel. $R^2$ and P value for linear regression reported in **g**,**n**. P value from chi-squared test reported with susceptibility ratios (**m**). Data are presented as mean ± s.e.m.

2m). Furthermore, JZL-184 treatment strengthened the correlation between baseline latency and post-stress reduction in latency observed in naturally resilient mice (naturally resilient $r^2 = 0.34$; total population after JZL-184 treatment $r^2 = 0.81$; Fig. 2g versus n). These data indicate that JZL-184 promotes the expression of a stress-resilient phenotype.

**2-AG augmentation converts susceptibility into resilience**. Although our data demonstrate that increasing 2-AG can prevent the emergence of stress-susceptibility at a population level, these results could be due to either a profound reduction in latencies specifically of mice that if untreated would have been susceptible to stress, or by an unbiased anxiolytic effect on the entire population. To distinguish between these possibilities, we determined the effect of systemic 2-AG augmentation in pre-identified stress-susceptible and resilient populations. To this end, we first established that stress-susceptibility was a relatively stable trait across at least two stress exposures. Extending our data above, we show that two stress exposures 1 week apart both significantly increase latency relative to baseline specifically in mice categorized as susceptible based on the first stress novel cage test (Fig. 3a–c). NIH novel-cage test consumption did not significantly change after or between stress exposures across the whole population or in either subpopulation (Fig. 3d–f). Figure 3g shows an explicit comparison between the latency change from baseline after the 1st and 2nd stress exposures; importantly, in the stress-susceptible subpopulation there was no significant habituation between the first and second stress-test exposures. While the difference trended downward from the 1st to the 2nd stress-test in both subpopulations, suggesting the possibility of a slight habituation, the decrease was not significant and the latency delta from baseline remained significantly different between the subpopulations after both stress exposures. These data also argue against statistical anomalies such as regression to the mean confounding subsequent drug effect studies (see below). To further exclude the possible confound of between-test habituation, we plotted 1st stress-test latency against the change in latency between the 1st and 2nd stress-tests for the susceptible mice and found that there was no correlation between degree of habituation and latency after the 1st stress exposure (Fig. 3h). In other words, the degree of stress-induced anxiety-like behaviour did not predict the magnitude of habituation to the second stress-test exposure. Lastly, we analysed the duration of behavioural dysregulation induced by acute foot-shock stress exposure. In stress-susceptible mice, feeding latency in the NIH assay decreases from 1 day to 3 days, and from 1 day to 14 days after stress exposure; but remains significantly higher at 3 days and 14 days after stress compared with baseline (Supplementary Fig. 3). Latencies for the stress-resilient subpopulation do not differ at any time point after stress (Supplementary Fig. 3).

These experiments indicate that stress-susceptibility is a relatively stable trait across at least two stress-NIH tests, which allows us to explicitly determine the effects of 2-AG augmentation on a priori defined stress-susceptible and resilient populations. Using this repeated stress NIH paradigm, we found that JZL-184 was able to reverse established stress-susceptibility. JZL-184 reduced latency and increased consumption after stress across the whole population and Rimonabant blocked these effects (Fig. 4a,b). To examine subpopulation-specific drug effects, mice were split into stress-resilient (Fig. 4c,d) and susceptible (Fig. 4e,f) subpopulations using the 120 s cutoff criterion established in Fig. 2. The paired baseline and post-stress individual latency and consumption data in Fig. 4d,f illustrate the disparate effect of stress on latency and consumption between resilient and susceptible mice, despite widespread overlap in

baseline latencies. JZL-184 reduced stress-test latency in the susceptible subpopulation without affecting latency in the resilient subpopulation (Fig. 4c,e). This stress-resilience promoting effect of JZL-184 in susceptible mice was completely blocked by co-treatment with Rimonabant. Interestingly, specifically in the resilient subpopulation, Rimonabant treatment increased latency significantly beyond stress-test latency (see Fig. 4c, NC-FS-V versus NC-FS-JZL-RIM). These data suggest the possibility that resilient mice have elevated cannabinoid signalling either at baseline or specifically in response to stress, which may be mediating their stress-resilience. Explicit comparison of susceptible and resilient latencies between the 1st stress test with vehicle treatment and the 2nd stress-test with JZL-184 demonstrated that JZL-184 reversed susceptibility to stress after it had manifested (Fig. 4g). Furthermore, the severity of stress-induced anxiety-like behaviour was significantly correlated with the effectiveness of JZL-184 treatment (Fig. 4h), such that larger stress-induced latency increases corresponded with larger JZL-184-induced latency reductions in the subsequent test. As depicted in Fig. 4i, these data indicate that increasing 2-AG-CB1R signalling converted a portion of the stress-susceptible subpopulation into stress-resilient mice. Although we have shown that ability of JZL-184 to reduce feeding latency after acute stress is not driven by increased appetite (Supplementary Fig. 1), we wanted to confirm this using our repeated NIH testing paradigm. Specifically, when stress-resilient and susceptible mice are tested in the home cage rather than novel cage after repeated stress, JZL-184 does not affect latency or consumption (Fig. 4j). If increased appetitive drive was responsible for the changes in feeding latency after repeated stress, increases in food intake should be observed under home-cage conditions. Moreover, if increased appetitive drive was responsible for the changes in feeding latency there should be an inverse correlation between novel-cage feeding latency and food consumption; however, we found no correlation between novel-cage latency and consumption further arguing against changes in appetite driving changes in latency measurements with vehicle or JZL-184 treatment (Fig. 4k). We were not able to detect an anxiolytic-like effect of JZL-184 in the EPM in the context of the repeated testing paradigm, however this may be due to the reduced sensitivity of this assay to 2-AG augmentation (Supplementary Fig. 4). Indeed, a recent study found that MAGL inhibition was not effective in reducing anxiety-like behaviour in the EPM in high anxiety rats[36].

Female mice displayed similar stress-susceptibility and JZL-184 efficacy (Supplementary Fig. 5). However, in this case baseline NIH consumption significantly differed between susceptible and resilient mice, and at a population level moderately predicted stress-induced changes in latency ($n = 51$, $r^2 = 0.12$, $P = 0.017$, linear regression; Supplementary Fig. 5). These data suggest possible sex differences in acute stress responsivity, which will require further investigation to validate.

**2-AG depletion converts resilience into susceptibility**. We next tested whether acute systemic pharmacological 2-AG depletion with the DAGLα inhibitor DO34 (ref. 37), which selectively reduces brain 2-AG and AA but not AEA levels (Fig. 5a–c), would increase susceptibility to stress-induced anxiety-like behaviour. As expected, at a population level DO34 administration increases feeding latency and decreases consumption after stress relative to vehicle treatment (Fig. 5d,e). Subgroup analysis revealed DO34 strongly increases feeding latency in previously resilient mice without significantly affecting latency in susceptible mice (Fig. 5f–h), and increases the proportion of stress-susceptible mice relative to vehicle treatment (Fig. 5i). Given the large reduction in consumption observed after DO34 treatment, we

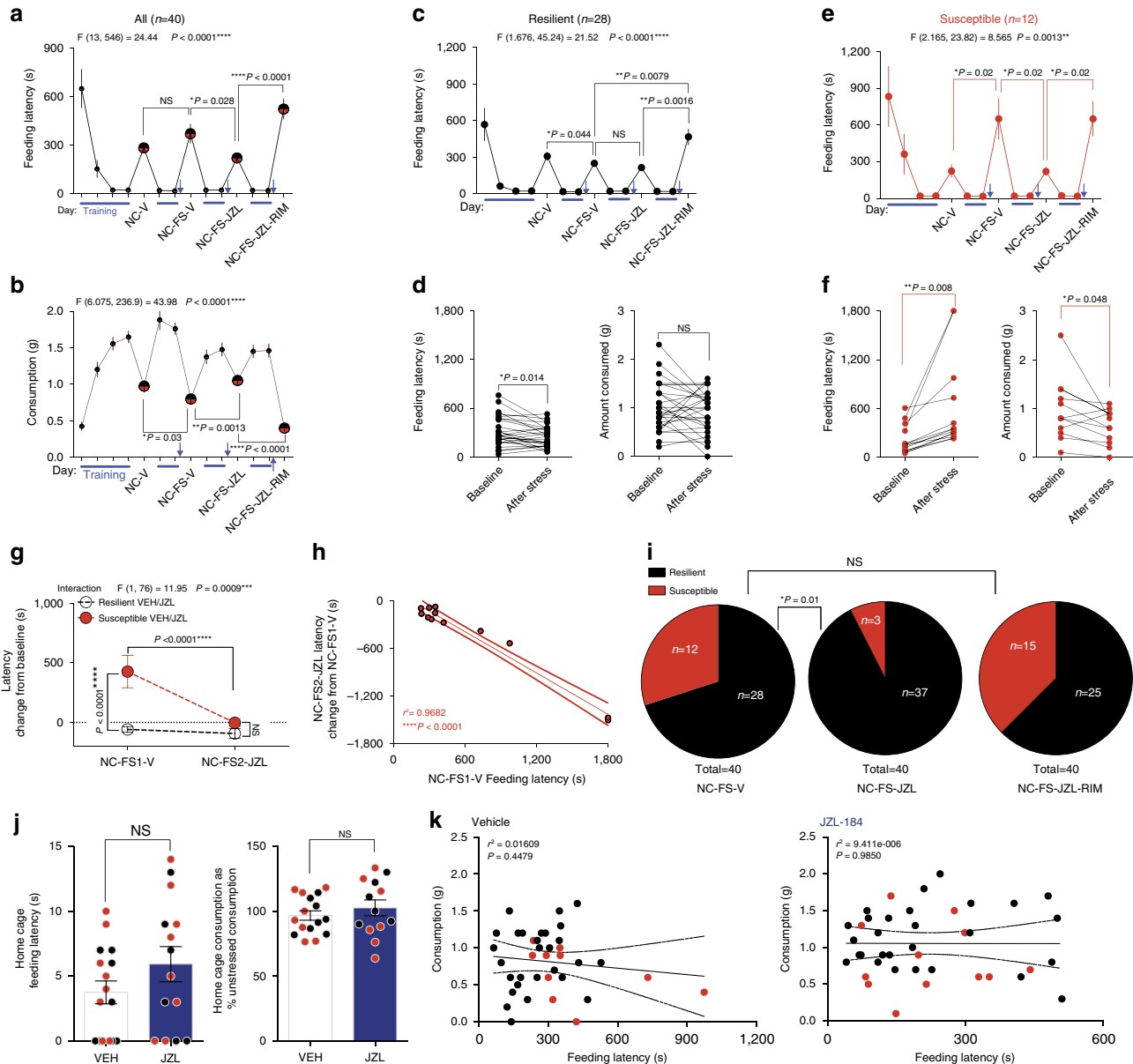

**Figure 4 | 2-AG augmentation promotes resilience to acute stress-induced anxiety-like behaviour.** (**a**) Home cage training (blue lines) and novel cage (NC) latencies and (**b**) consumption before (NC-V) and after (NC-FS-V) foot-shock stress with vehicle (V), JZL-184, and JZL-184 + Rimonabant (RIM) treatment. (**c**) Resilient subgroup latencies separated from **a**. (**d**) Resilient individuals' baseline and post-stress latencies and consumption. (**e**) Susceptible subgroup latencies separated from **a**. (**f**) Susceptible individuals' baseline and post-stress latencies and consumption. (**g**) Direct comparison of changes in latency from baseline between the 2nd stress test with JZL-184 (NC-FS2-JZL) and the first stress test with vehicle (NC-FS1-V). (**h**) Correlation between stress-test latency and change in latency between JZL-184 and vehicle treatment for susceptible individuals. (**i**) Stress-susceptibility ratios for the same cohort of mice across three weeks after vehicle, JZL-184, or JZL-184 + Rimonabant treatment. (**j**) Home cage testing latency (left) and consumption as % of previous day's home cage/no stress consumption (right) 24 h after stress exposure with resilient (black circles) and susceptible (red circles) individuals treated with vehicle (VEH) or JZL-184 (blue) one week after susceptibility categorization. (**k**) Whole population correlations between post-stress novel cage test feeding latency and consumption with vehicle and JZL-184 treatment. Blue arrows indicate foot-shock stress exposure. F and P values for one-way (**a–c,e**) or two-way (**g**) ANOVA shown above individual panels. P values for pairwise comparisons derived from Holm-Sidak multiple comparisons test after ANOVA, unpaired two-tailed t-test (**j**), or paired two-tailed t-test (**d,f**) shown in panels. $R^2$ and P value for linear regression shown in **h,k**. P values from chi-squared tests reported in **i**. Data are presented as mean ± s.e.m.

wanted to determine if the anxiogenic-like effect (increased latency) was dependent on changes in appetitive drive. We found that feeding latency and consumption in novel-cage testing after stress were not correlated in either vehicle-treated or DO34-treated resilient groups, again indicating that these two measures are independent (Fig. 5j). All individuals that did not drink during the entire testing period were excluded from this analysis,

although the linear regression remains insignificant even when they are included. We also tested DO34 in a secondary assay, the EPM, to confirm its anxiogenic effect. Using the same experimental design, with the exception that mice were tested in the EPM rather than in the NIH assay in week 3, 24 h after the 2nd stress exposure, we found resilient mice treated with DO34 showed higher levels of anxiety-like behaviour in the EPM

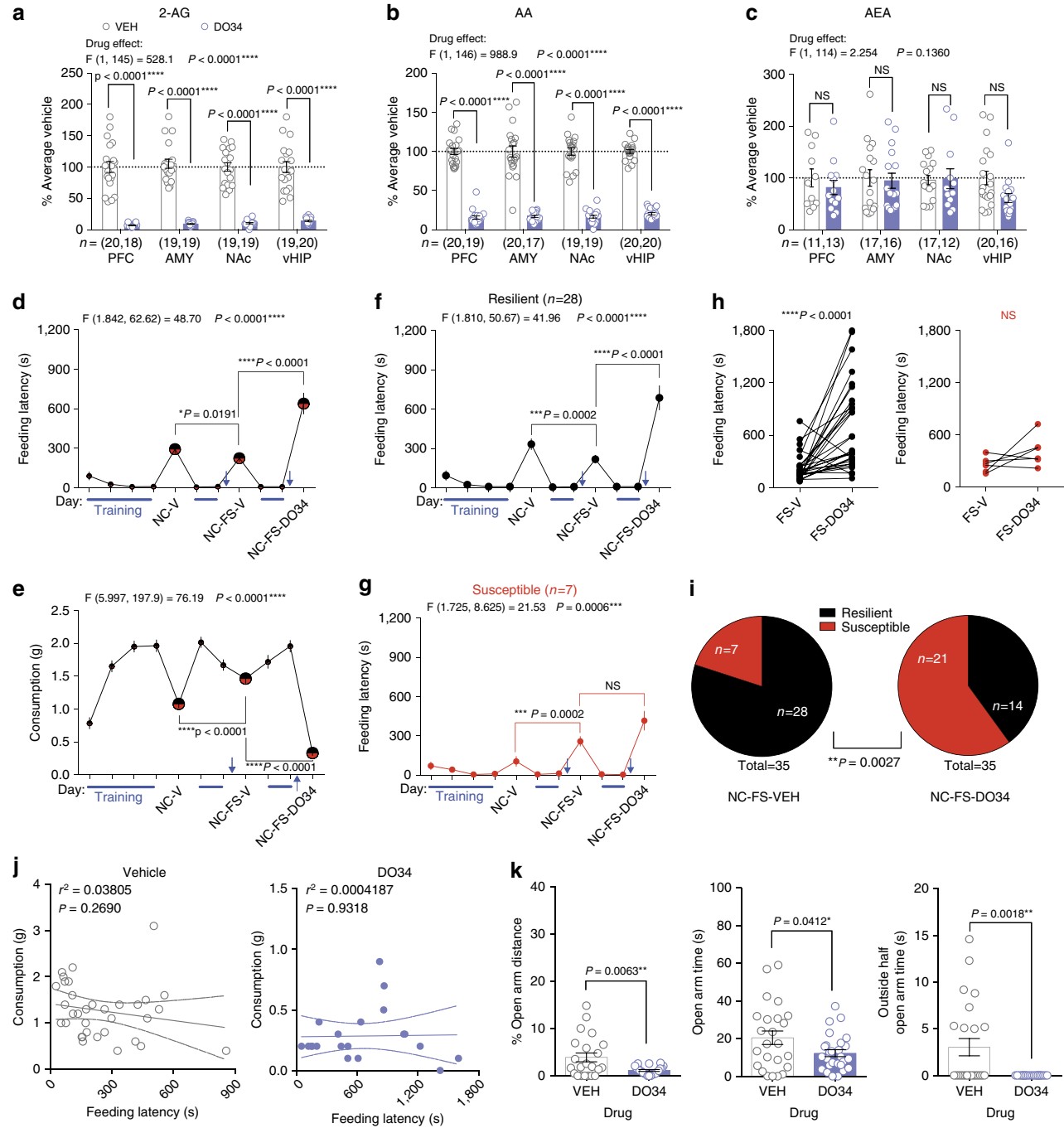

**Figure 5 | 2-AG depletion increases susceptibility to acute stress-induced anxiety-like behaviour.** (**a–c**) Effects of DO34 (50 mg kg$^{-1}$; purple) on 2-arachidonoylglycerol (2-AG), arachidonic acid (AA), and anandamide (AEA) in the prefrontal cortex (PFC), amygdala (AMY), nucleus accumbens (NAc), and ventral hippocampus (vHIP). (**d**) Home cage training (blue lines) and novel cage (NC) latencies and (**e**) consumption before (NC-V) and after (NC-FS-V) foot-shock stress with vehicle (V) or DO34 treatment. (**f**) Resilient and (**g**) susceptible subgroup latencies separated from **d**. (**h**) Effects of DO34 on feeding latency relative to vehicle treatment in resilient (black) and susceptible (red) mice. (**i**) Stress-susceptibility ratios for the same cohort of mice across two weeks after vehicle or DO34 treatment. (**j**) Resilient population correlations between novel cage test feeding latency and consumption with vehicle and DO34 treatment. (**k**) Elevated plus maze 24 h after stress exposure with resilient individuals treated with vehicle (VEH) or DO34 (purple) one week after susceptibility categorization. Blue arrows indicate foot-shock stress exposure. F and P values for one-way (**d–g**) or two-way (**a–c**) ANOVA shown above individual panels. P values for pairwise comparisons derived from Holm-Sidak multiple comparisons test after ANOVA, unpaired two-tailed t-test (**k**), or paired two-tailed t-test (**h**) shown in panels. R$^2$ and P value for linear regression shown in **j**. P value from chi-squared test reported in **i**. Data are presented as mean ± s.e.m.

relative to vehicle-treated resilient mice (Fig. 5k). Taken together, these data indicate that acute depletion of 2-AG signalling increases susceptibility to the adverse behavioural effects of traumatic stress exposure.

**2-AG-CB1 signalling components do not vary with resilience.** In an initial effort to elucidate the eCB-related biochemical mechanisms contributing to stress-susceptibility, we analysed levels of CB1, DAGLα and MAGL in multiple limbic brain

regions. We found no significant differences in levels of CB1, DAGLα, or MAGL (Supplementary Fig. 6) protein in the amygdala, prefrontal cortex (PFC), nucleus accumbens (NAc), or

ventral hippocampus (vHIP) of resilient or susceptible mice. We next used mass spectrometry to directly measure the bulk tissue levels of 2-AG in these brain regions. We found no significant

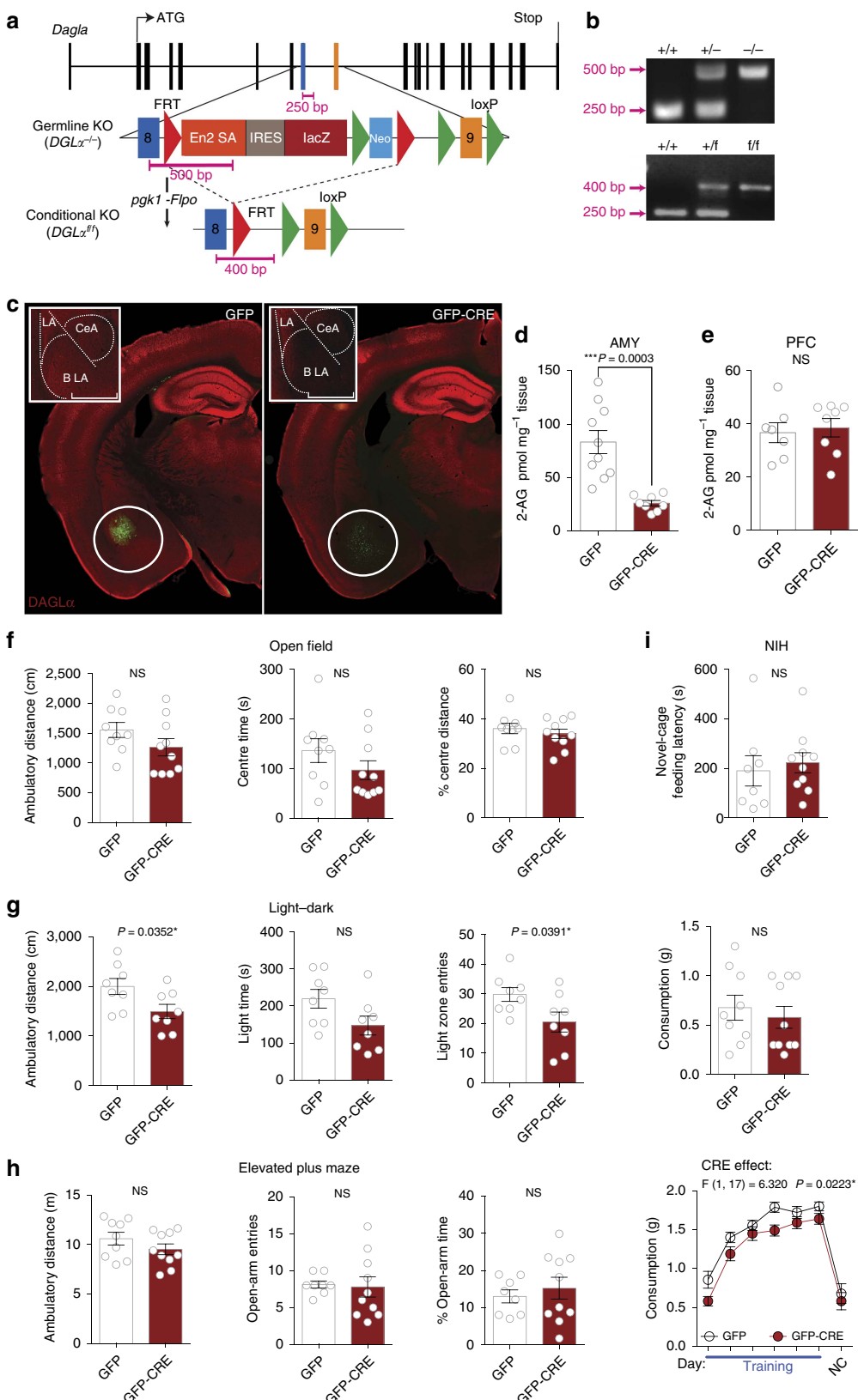

differences in the levels of 2-AG in the amygdala and furthermore show that amygdala 2-AG does not correlate with stress-induced change in latency across the whole population, or specifically in the resilient, or susceptible subpopulations (Supplementary Fig. 7). We further found no group differences in 2-AG levels, or population or subpopulation correlations with susceptibility in the PFC, NAc, or vHIP (Supplementary Fig. 7). Although bulk 2-AG levels and total protein levels of 2-AG metabolic enzymes were not different between stress-susceptible and resilient mice, several post-translational mechanisms of regulation have been discovered that may affect the efficiency of 2-AG signalling, including DAGLα phosphorylation[38] or localization[39], and MAGL sulfenylation[40]. Future studies will be required to test mechanisms by which 2-AG signalling could be differentially regulated in stress-resilient versus susceptible mice.

**Tetrahydrocannabinol promotes stress-resilience.** Given the high prevalence of cannabis use in patients with anxiety disorders and PTSD, and the high rate of symptom-coping motives cited by cannabis users with these disorders[41,42], we next sought to determine if this resilience-promoting effect of 2-AG was generalizable to a direct CB1R agonist. Consistent with clinical reports, we found that a low dose of the cannabinoid agonist, delta-9-tetrahydrocannabinol (THC, $0.25 \, mg \, kg^{-1}$), also specifically reduced stress-induced anxiety-like behaviour in stress-susceptible mice without affecting consumption in either group (Supplementary Fig. 8). These data are consistent with several reports showing direct activation of CB1 receptors can attenuate adverse consequences of traumatic stress in laboratory models[43,44], and provide additional preclinical support for the tension-reduction hypothesis to explain high rates of cannabis use in patients with mood and anxiety disorders.

**Amygdala 2-AG is necessary for adaptation to repeated stress.** To dissect region-specific necessity of 2-AG signalling in the regulation of stress susceptibility and adaptation to repeated stress exposure, we developed a floxed mouse line for conditional Cre-dependent deletion of the primary central 2-AG synthetic enzyme DAGLα (DAGLα$^{f/f}$) (Fig. 6a,b and see Methods). We utilized stereotaxic injection of adeno-associated virus serotype 5 encoding a green fluorescent Cre recombinase fusion protein (AAV-GFP-CRE) into the basolateral amygdala (BLA), PFC and nucleus accumbens (NAc) of DAGLα$^{f/f}$ mice to achieve substantial reductions in DAGLα immunoreactivity compared with AAV-GFP control injection (Fig. 6c). As expected, BLA injections of AAV-GFP-CRE also produced a significant reduction of amygdala 2-AG levels (Fig. 6d), but had no effect in a non-injection region, the PFC (Fig. 6e). BLA AAV-GFP-CRE injection did not affect locomotor or basal anxiety-like behaviour in the OFT (Fig. 6f). AAV-GFP-CRE injected mice did exhibit a slight basal anxiety-like phenotype in the light-dark test (Fig. 6g), but not in EPM or NIH (Fig. 6h,i). BLA-specific DAGLα deletion therefore produced only a slight basal anxiety-like phenotype, in stark contrast to germline deletion[16,30]. DAGLα deletion within the PFC or NAc did not affect basal anxiety-like behaviours (Supplementary Fig. 9).

We next used the repeated NIH paradigm to test the hypothesis that BLA-specific deletion of DAGLα increases stress-susceptibility. We again found that BLA AAV-GFP-CRE mice exhibited baseline NIH latencies comparable to BLA AAV-GFP injected mice, and additionally that group latencies did not diverge after single stress exposure (Fig. 7a). However, as others have shown, 2-AG progressively increases in response to repeated homotypic stress exposure[31,32]. This progressive increase in 2-AG after repeated homotypic stress has been suggested to represent part of the endogenous stress adaptation response[31]. Based on these data, and our previous experiments showing JZL-184 decreases anxiety-like behaviours after repeated foot-shock stress exposure (Fig. 1), we hypothesized that BLA 2-AG signalling may become increasingly important in mediating adaptation across repeated stress exposures. Consistent with this hypothesis, 24 h after a 5th foot-shock stress exposure, BLA AAV-GFP-CRE mice exhibited significantly higher NIH latencies than BLA AAV-GFP mice (Fig. 7a; NC-5FS). Applying the 120 s cutoff criterion to the 5-day stress latencies, we split the AAV-GFP and AAV-GFP-CRE mice into resilient and susceptible subpopulations (Fig. 7b,c). A significantly larger proportion of AAV-GFP-CRE mice exhibited susceptibility after 5 days of stress (Fig. 7d) and, furthermore, feeding latency was significantly higher in susceptible BLA AAV-GFP-CRE mice than susceptible BLA AAV-GFP mice, suggesting an increased severity of the stress-induced anxiety-like phenotype (Fig. 7e,f). These data indicate that BLA 2-AG signalling is necessary for the physiological adaptation to repeated homotypic stress, and that other regions or the coordinated actions of 2-AG signalling within multiple brain regions promote resilience in response to acute stress exposure. In contrast, mice with PFC or NAc DAGLα deletion did not exhibit differential susceptibility to acute stress-induced anxiety-like behaviour or adaptation to repeated stress, relative to GFP-injected controls (Supplementary Fig. 10).

**Resilience is associated with enhanced BLA 2-AG signalling.** To elucidate synaptic and circuit-level mechanisms by which 2-AG signalling promotes stress-resilience, we determined the synaptic efficacy of 2-AG signalling in resilient and susceptible subpopulations using electrophysiological approaches. We utilized *ex vivo* whole-cell patch-clamp electrophysiology to examine the effect of JZL-184 incubation on the frequency and amplitude of spontaneous excitatory postsynaptic currents (sEPSCs) onto pyramidal cells in the BLA of control and stress-exposed mice. As expected, in unstressed mice JZL-184 significantly reduced sEPSC frequency (increased inter-event interval; IEI) without affecting sEPSC amplitude (Fig. 8a). Interestingly, one day after stress, JZL-184 significantly reduced both sEPSC frequency and amplitude

---

**Figure 6 | Conditional DAGLα knockout mice and BLA-specific DAGLα deletion.** (**a**) Diagram of targeting construct and strategy for the generation of DAGLα$^{f/f}$ mouse. Mice harboring *dagla* gene-trap cassette[16] were crossed to *pgk-Flpo* mice to generate conditional knockouts with loxP sites flanking exon 9. (**b**) PCR products for genotyping of germline (DAGLα$^{-/-}$) and conditional (DAGLα$^{f/f}$) knockouts. Primer binding sites shown in **a**. (**c**) Representative coronal brain slices from DAGLα$^{f/f}$ mouse after BLA-AAV-GFP (left) and BLA-AAV-GFP-CRE (right) injection, and 20X magnification of BLA-DAGLα immunoreactivity of BLA-GFP control and BLA-GFP-CRE injected mice (square insets). White circles represent typical brain punch dissections for mass spectrometry. Inset scale bars are 500 μm. (**d**) Amygdala 2-AG levels after AAV-GFP and AAV-GFP-CRE BLA-injection from punch biopsies as indicated by white circles in **c**. (**e**) PFC 2-AG levels after BLA-AAV-GFP and BLA-AAV-GFP-CRE injection. (**f**) Effect of AAV-GFP vs. AAV-GFP-CRE BLA-injection on behaviour in open-field, (**g**) light-dark box, and (**h**) elevated plus-maze. (**i**) Effect of AAV-GFP vs. AAV-GFP-CRE BLA-injection on baseline novelty-induced hypophagia (NIH) testing. *P* values shown for unpaired one-tailed *t*-test above each **d-i**. F and *P* values for two-way ANOVA shown in **i**. Data are presented as mean ± s.e.m.

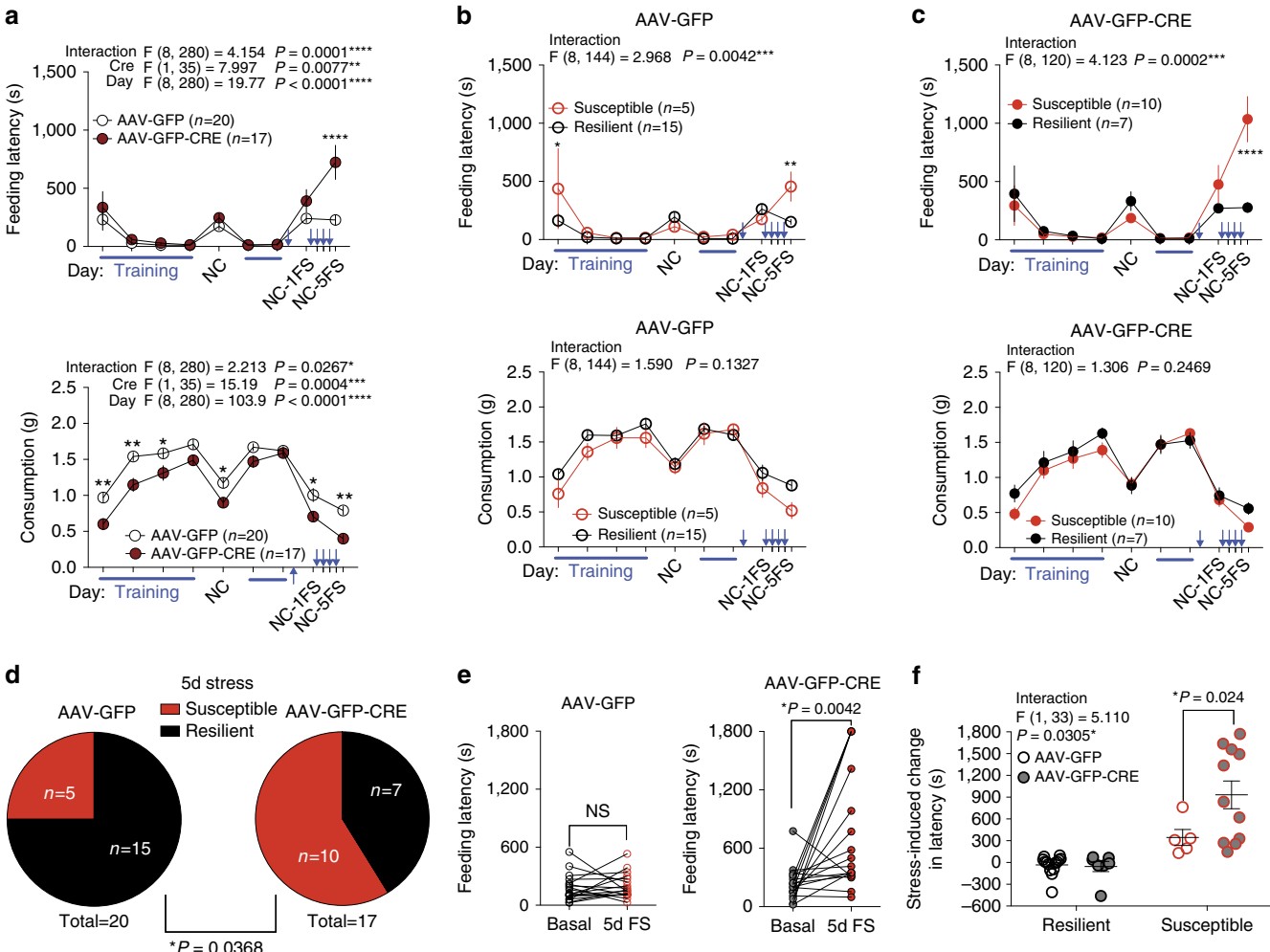

**Figure 7 | BLA-2-AG signalling is required for resilience to repeated traumatic stress.** (**a**) Effect of AAV-GFP vs. AAV-GFP-CRE BLA-injection on home cage NIH training (blue lines) and novel cage latency (top) and consumption (bottom) with no stress (NC), 24 h after 1 day of stress (NC-1FS), and 24 h after a 5th day of stress (NC-5FS). (**b**) AAV-GFP latency (top) and consumption (bottom) from **a** split into resilient (black) and susceptible (red) groups. (**c**) AAV-GFP-CRE latency (top) and consumption (bottom) from **a** split into resilient (black) and susceptible (red) groups. (**d**) 5-day stress susceptibility ratios for BLA AAV-GFP and AAV-GFP-CRE injected groups. (**e**) Paired individual baseline and post-stress latencies in AAV-GFP and AAV-GFP-CRE BLA-injected groups. (**f**) Direct comparison of stress-induced changes in latency in resilient and susceptible AAV-GFP vs. AAV-GFP-CRE BLA-injected mice. Data combined from 2 independent cohorts. Blue arrows in **a–c** indicate stress exposure, which occurred once per day for 5 consecutive days. F and P values for two-way ANOVA shown above individual (**a–c,f**). P values for pairwise comparisons derived from Holm-Sidak multiple comparisons test after ANOVA, paired two-tailed t-test (**e**), and chi-squared test reported (**d**) in each panel. Data are presented as mean ± s.e.m.

(Fig. 8b). Direct comparison of all four conditions revealed that stress increased sEPSC frequency, and that JZL-184 was more effective at reducing sEPSC frequency after stress (Fig. 8c), suggesting 2-AG signalling limits stress-related excitatory drive to BLA neurons.

We next examined potential differences in tonic 2-AG signalling at BLA glutamatergic synapses in resilient versus susceptible mice after stress. JZL-184 significantly reduced sEPSC frequency 24 h after stress exposure in both populations while Rimonabant increased sEPSC frequency in the resilient subpopulation only; no differences in sEPSC frequency were observed between vehicle-treated susceptible and resilient mice (Fig. 8d). Neither drug significantly affected sEPSC amplitudes (Fig. 8d, bottom). Direct comparison of sEPSC frequencies and amplitudes with maximal (JZL-184 incubated) and abolished (Rimonabant incubated) 2-AG signalling revealed that stress-resilience was associated with a greater range of 2-AG signalling capacity at BLA glutamatergic synapses (Fig. 8e, representative

traces in 8f). These data suggest stress-resilient mice utilize eCB signalling to regulate BLA glutamatergic transmission in a wider dynamic range than stress-susceptible mice, and that this broader utilization of eCB signalling could represent a synaptic substrate promoting stress-resilience.

**2-AG modulation of vHIP-BLA inputs is enhanced in resilience.** Our electrophysiological data thus far indicate enhanced 2-AG signalling at BLA glutamatergic synapses in stress-resilient mice; however, the BLA receives a variety of glutamatergic afferents. Indeed, optogenetic projection-targeting approaches combined with *ex vivo* electrophysiology revealed glutamatergic inputs from the PFC, vHIP and lateral entorhinal cortex (Supplementary Fig. 11). Of these afferents, the vHIP input showed the largest sensitivity to phasic 2-AG-mediated retrograde inhibition in the form of optogenetic depolarization-induced suppression of excitation (oDSE, Supplementary Fig. 11).

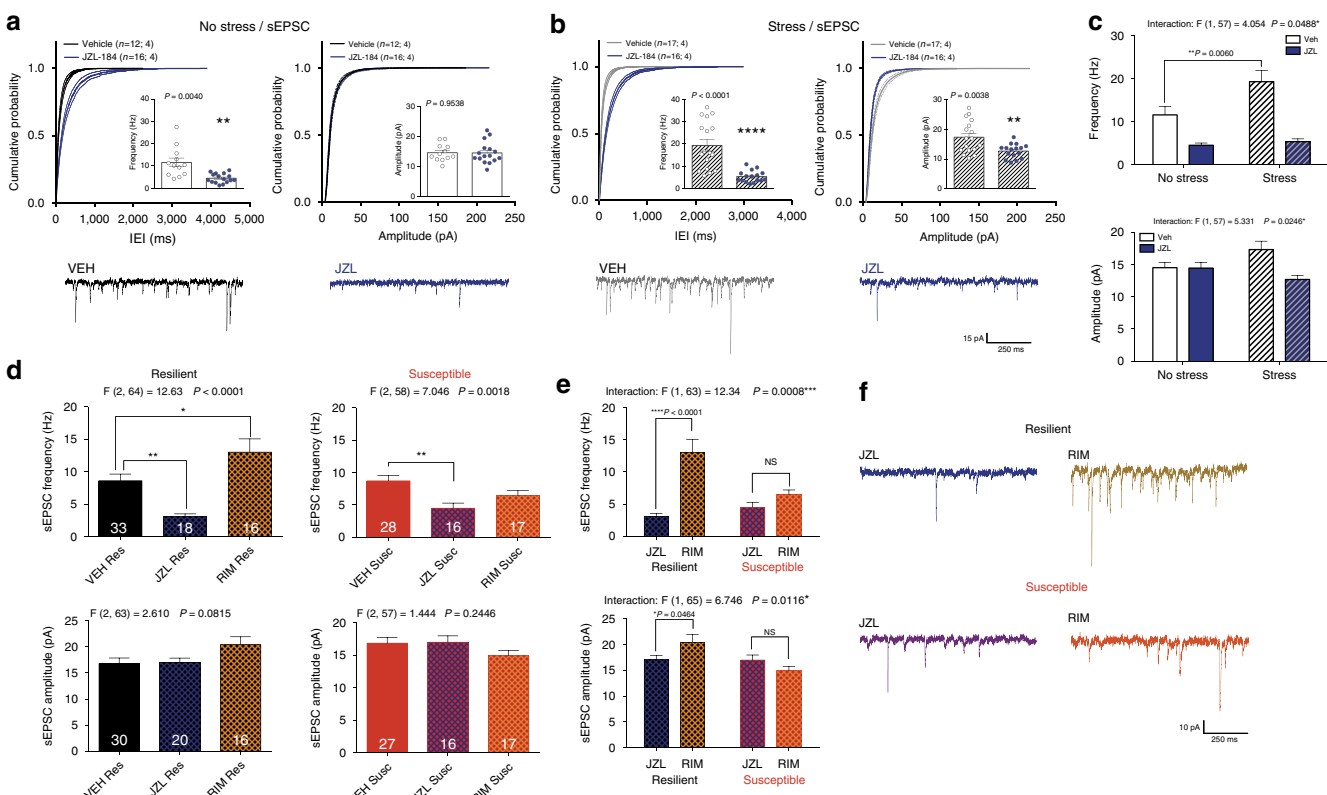

**Figure 8 | Stress-induced increases in sEPSC frequency in the BLA are eliminated by JZL-184 incubation.** (**a**) Effect of JZL-184 incubation on the inter-event interval (IEI; left), frequency (left inset) and amplitude (right) of spontaneous excitatory postsynaptic currents (sEPSCs) onto BLA pyramidal cells in control non-stressed mice, and (**b**) 24 h after foot-shock stress exposure. (**c**) Direct comparison of stress effect and JZL-184 effect from **a** and **b**. (**d**) Direct comparison of resilient (left; black) and susceptible (right; red) BLA sEPSC frequency (top) and amplitude (bottom) with vehicle (VEH) and either JZL-184 (blue hash) or Rimonabant (RIM; orange hash) incubation. (**e**) Direct comparison of the dynamic range of eCB signalling (defined as the difference in sEPSC frequency (top) and amplitude (bottom) between JZL-184 and Rimonabant conditions shown in **d** with representative traces in **f**. Number of cells is shown for each group. Number of (cells; animals) are reported in **a** and **b**. Number of cells reported in **d**. F and P values for one-way (**d**) or two-way (**c**,**e**) ANOVA shown above individual panels. P value for pairwise comparisons derived from Holm-Sidak multiple comparison test after ANOVA (**c**–**e**) or unpaired t-test (**a**,**b**) shown in individual panels. Data are presented as mean ± s.e.m.

We further verified that the vHIP-BLA oDSE was CB1-dependent by blocking it with Rimonabant (Supplementary Fig. 11). Given that the NIH test relies on perceived novelty of environmental context, and the vHIP-BLA circuit is considered to be anxiogenic[45,46], we next focused on elucidating the role of 2-AG signalling at these synapses in stress-resilient and susceptible mice.

To determine whether stress-resilience was associated with relatively enhanced 2-AG-mediated modulation of vHIP-BLA circuits, AAV-ChR2 was injected into the ventral hippocampus, followed by behavioural separation of mice into stress-susceptible and resilient populations, and ex vivo electrophysiological recordings (Fig. 9a, schematic). Input–output curves indicate that vHIP-BLA connectivity is stronger in stress-susceptible, relative to resilient, mice while paired pulse ratios do not differ between groups (Fig. 9b). Importantly, oDSE at vHIP-BLA synapses is significantly reduced in stress-susceptible, relative to resilient, mice (Fig. 9c). Moreover, JZL-184 restores maximal oDSE in stress-susceptible mice to levels seen in stress-resilient mice (Fig. 9d). In a subset of cells, we show that 1 μM JZL-184 wash-on specifically enhances oDSE in the susceptible group as demonstrated by paired pre versus post JZL-184 maximal oDSE measurements (Fig. 9e). However, JZL-184 wash-on does not differentially affect tonic 2-AG mediated oEPSC depression in stress-resilient versus susceptible groups (Fig. 9f). Injection site and fiber innervation of the BLA is shown in Fig. 9g. These data

reveal enhanced phasic, but not tonic, 2-AG signalling in stress-resilient, relative to susceptible, mice. These studies parallel our behavioural findings and indicate stress-resilient mice have both weaker connectivity of the anxiogenic vHIP-BLA pathway as well as elevated phasic 2-AG-mediated suppression of vHIP-BLA glutamatergic transmission. This elevated phasic 2-AG signalling likely contributes to their stress-resilient phenotype and the anxiogenic effects induced by CB1R blockade in these mice as CB1 receptor function at vHIP-BLA synapses as measured by synaptic depression induced by the direct CB1 agonist CP55940 does not differ between resilient and susceptible individuals (Supplementary Fig. 12). In contrast, stress-susceptible mice have relatively reduced phasic 2-AG signalling at vHIP-BLA synapses, which could contribute to their behavioural stress-susceptibility; both of which can be normalized by JZL-184 treatment.

In contrast to the vHIP-BLA pathway, we do not find any changes in synaptic connectivity between stress-susceptible and resilient mice at PFC-BLA synapses (Fig. 9h). Moreover, oDSE and tonic 2-AG-mediated suppression of glutamate release at PFC-BLA synapses are not different between stress-susceptible and resilient mice (Fig. 9i–l), and JZL-184 enhances oDSE comparably in both populations (Fig. 9j). Injection site and fiber innervation of the BLA are shown in Fig. 9m. These data support a degree of specificity for eCB effects within the vHIP-BLA circuit in the regulation of stress resilience.

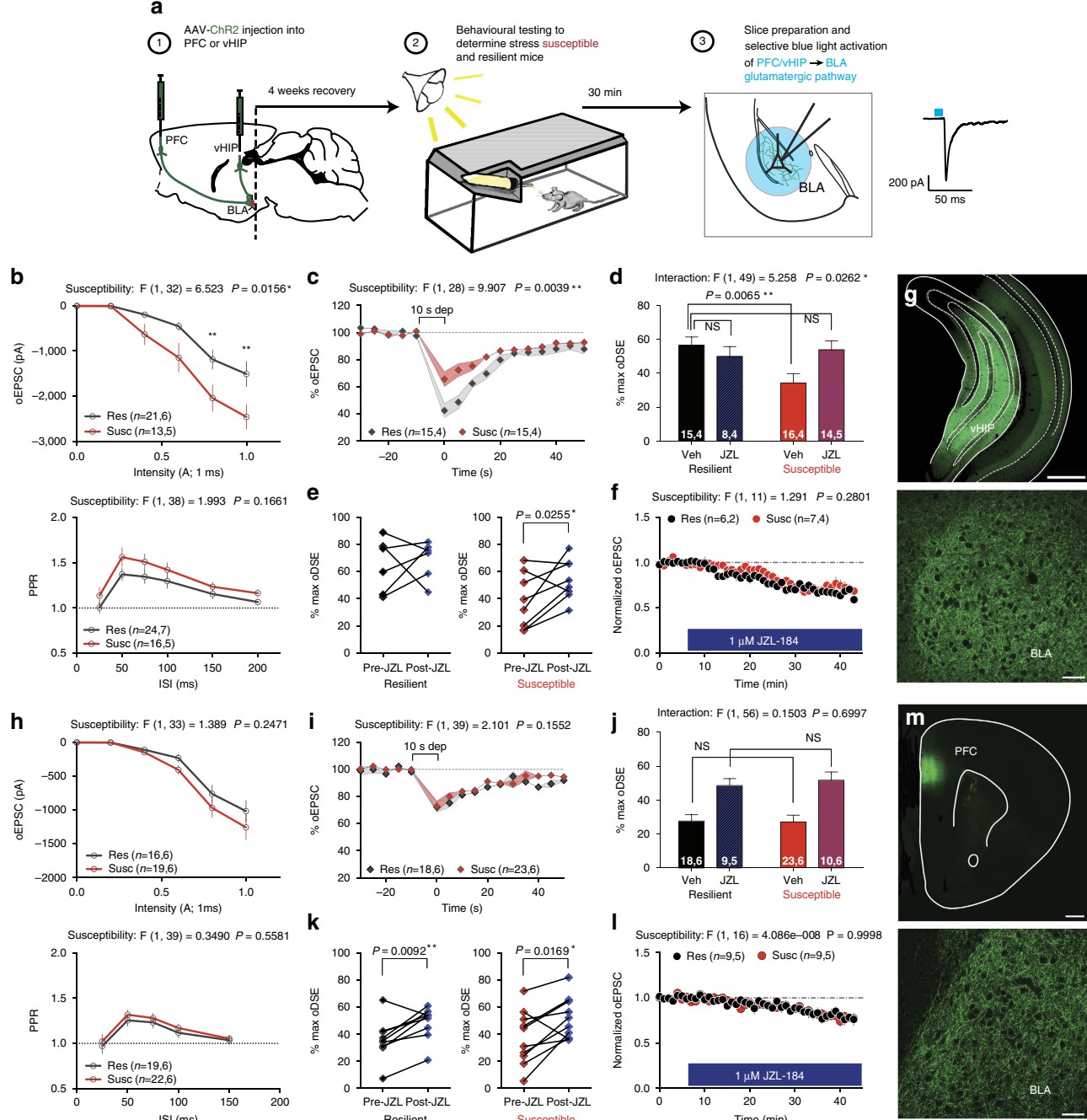

**Figure 9 | Stress-resilience is associated with greater 2-AG modulation of vHIP-BLA glutamatergic synapses.** (**a**) Schematic of the experimental procedure for optogenetic recordings. (**b–g**) Optogenetic recordings at vHIP-BLA synapses. (**h–m**) Optogenetic recordings at PFC-BLA synapses. (**b,h**) Optically evoked input-output curves and paired pulse ratio. (**c,i**) Depolarization-induced suppression of excitation (oDSE) in susceptible and resilient population. (**d,j**) Direct comparison of the effect of JZL-184 on the magnitude of oDSE in resilient versus susceptible groups. (**e,k**) Paired comparison of % maximal oDSE in the same cell pre- and post-JZL-184 incubation for resilient and susceptible groups. (**f,l**) 1 μM JZL-184-induced depression of optically evoked EPSCs. (**g,m**) Representative images of the vHIP and PFC injection sites, and the corresponding BLA recording sites. Number of (cells, animals) are presented within each panel. F and P values for two-way ANOVA shown above individual (**b–d,f,h–j,l**). P values shown for pairwise comparisons derived from Holm-Sidak multiple comparisons test after ANOVA or paired two-tailed t-test (**e,k**). Data are presented as mean ± s.e.m. Scale bars are 500 μm for vHIP and PFC images, 50 μm for BLA.

## Discussion

Given the prominent role of stress in the development and exacerbation of affective disorders including major depression and PTSD[1,2,4,7], understanding the biological mechanisms contributing to inter-individual differences in stress-susceptibility could lead to the development of susceptibility biomarkers, novel treatments, and preventative approaches[8,9]. Here we provide converging pharmacological, physiological, and genetic evidence supporting increased 2-AG-CB1R signalling as an endogenous stress-resilience factor that buffers against adverse consequences of stress. These data further support pharmacological 2-AG augmentation as a viable approach for the treatment of stress-related neuropsychiatric disorders.

Utilizing individual differences in stress responsivity to elucidate biological mechanisms subserving stress-resilience/susceptibility has been of increasing interest in recent years. For example, using chronic social defeat stress (CSDS), Krishnan *et al.* showed that stress-resilience is an active process of adaptation associated with a multitude of changes in gene expression and neural signalling rather than merely an absence of maladaptive changes induced by stress[13]. While social interaction measures after CSDS distinguish subpopulations expressing anhedonia and other depressive-like phenotypes, equivalent levels of anxiety-like behaviour are observed in susceptible and resilient groups[13]. The learned helplessness model of depression has also produced important insights about the mechanisms underlying differential susceptibility to the development of depressive-like phenotypes at both synaptic and circuit levels[47–49]. Here we developed a novel model combining traditional conditioned fear-training (foot-shock + cue/context exposure) and the well-established NIH test of anxiety[35] to examine individual differences in the generalized anxiety-like response to stress. Overall ~1/3 of mice show a susceptible phenotype, defined as a ≥120 s increase in feeding latency 24 h after foot-shock stress relative to baseline latency. It may be important to note that testing was performed on young adult mice and it is possible that stress responsivity could differ in either younger or older populations. The stress-susceptible subpopulation defined using this approach also exhibited increased anxiety-like behaviour in the EPM and OFT, further validating our paradigm, without showing changes in depressive-like measures (not shown). Importantly, while the key phenotype in our model has almost completely decayed two weeks following stress exposure it consistently renews following an additional stress exposure which provides a unique opportunity to examine drug effects and manipulate susceptibility in pre-defined populations. Lastly, consistent with the CSDS model[13], susceptibility in our assay appears to be a latent trait, as neither baseline NIH latencies nor anxiety-like behaviour in the EPM or OFT differed between groups before stress exposure.

Given the emerging role of 2-AG signalling in the regulation of anxiety and stress-responses[30], we utilized this novel behavioural paradigm to test the hypothesis that 2-AG promotes resilience to stress. We show that acute systemic 2-AG augmentation robustly increases the proportion of mice exhibiting resilience to adverse consequences of acute stress, and promotes resilience in previously susceptible mice. In contrast, acute systemic pharmacological 2-AG depletion and CB1R blockade render previously stress-resilient mice susceptible to the development of anxiety-like behaviour after acute stress exposure. Taken together, these data provide causal evidence that 2-AG-CB1 signalling promotes resilience to the adverse effects of acute traumatic stress exposure.

Another important aspect of stress responsivity is habituation or adaptation to repeated homotypic stress exposure, which we have previously suggested may involve 2-AG signalling. Consistent with our previous hypotheses, we found that BLA-specific DAGLα deletion significantly impairs adaptation to repeated stress. Although BLA-specific DAGLα deletion minimally impacts baseline anxiety-like behaviours, it increases the proportion of mice showing anxiety-like behaviour after repeated stress exposure and the severity of the anxiety-like phenotype. The preferential effect of BLA-deletion of DAGLα after repeated stress is somewhat surprising in light of the strong converging pharmacological data indicating 2-AG-CB1R modulation of resilience to acute traumatic stress exposure. However, consistent with previous work in other stress models[50–52], it is likely that endogenous 2-AG plays a more important role in stress modulation after multiple homotypic stress exposures, thus region-specific loss-of-function manipulations may show larger effects after repeated stress. Alternatively, other brain regions besides those tested may be more critical for the modulation of acute stress-induced anxiety or compensatory changes could occur in the weeks after DAGLα deletion that counteract potential increases in acute stress-susceptibility. Despite this issue, these data overall are consistent with previous clinical and preclinical data suggesting 2-AG deficiency could contribute to the development of stress-related psychiatric disorders[16,53–55].

A primary function of 2-AG signalling is the retrograde synaptic suppression of afferent neurotransmitter release within limbic nodes including the amygdala and PFC[15,19,56,57]. While dysfunction of multiple limbic regions has been implicated in stress-related psychiatric disorders, hyperactivity of the amygdala, in particular, has been highly associated with affective disorders[58–61], and stress increases BLA neuronal activity in rodents[62,63]. Interestingly, the anxiolytic effect of low-dose cannabinoid agonist treatment is mediated through CB1Rs on forebrain glutamatergic, but not GABAergic, terminals[64,65], and deletion of CB1Rs from forebrain glutamatergic terminals produces increased fear behaviours[66]. Together these data suggest that pharmacological 2-AG augmentation may exert its anxiolytic and resilience-promoting effects by reducing BLA glutamatergic transmission. Indeed, we show that resilient mice have a larger difference in BLA sEPSC frequency between maximal 2-AG-CB1R signalling (JZL-184 incubated) and abolished 2-AG-CB1R signalling (Rimonabant incubated), compared with susceptible mice. These data suggest resilient mice utilize 2-AG signalling to regulate BLA afferent glutamatergic transmission within a broader dynamic range. We suggest this broader dynamic range of 2-AG-signalling could represent part of the adaptive response to traumatic stress that characterizes stress-resilience.

Importantly, we also find that stress resilience is associated with enhanced phasic, but not tonic, 2-AG-mediated suppression of glutamatergic transmission at vHIP-BLA synapses, and that relatively impaired 2-AG signalling at these synapses in stress-susceptible mice can be normalized by JZL-184 treatment. These data suggest 2-AG signalling could serve to reduce vHIP-BLA circuit activity to promote resilience and successful adaptation to stress. These data are consistent with the known role of the vHIP in relaying contextual information to limbic output structures to generate appropriate behavioural responses to changes in environmental context, and with the anxiogenic function of vHIP-BLA circuits[67].

Our data clearly show that systemic 2-AG augmentation promotes resilience to stress, and that stress-susceptible mice have relatively impaired synaptic 2-AG signalling at vHIP-BLA glutamatergic synapses which can be normalized by JZL-184 application *ex vivo*. However, these data do not conclusively localize the resilience-promoting effects of systemic 2-AG augmentation to reductions in activity of the vHIP-BLA circuit per se. Future studies utilizing BLA-specific 2-AG augmentation combined with vHIP-BLA pathway-specific CB1 deletion will be required to conclusively test this hypothesis. The involvement of other neural circuits known to regulate stress-responsivity and anxiety-related behaviours has also not been investigated in the current work.

In summary, here we utilize individual differences in stress-responsivity to demonstrate a causal role for 2-AG-mediated eCB signalling in promoting stress-resilience. Our data demonstrate that on both group (susceptible versus resilient) and individual (correlational analyses) levels, the severity of stress-induced generalized anxiety-like behaviour predicts the beneficial response to pharmacological 2-AG augmentation. We also find that stress-resilience is associated with enhanced 2-AG-CB1R mediated synaptic suppression of vHIP-BLA glutamatergic

transmission, and that BLA 2-AG signalling is required for successful adaptation to repeated traumatic stress. Altogether, these data suggest that pharmacological augmentation of 2-AG signalling could represent a novel approach for the treatment of stress-related neuropsychiatric disorders[20,24,68], and that 2-AG deficiency states could represent a stress-susceptibility endophenotype predisposing to the development of affective pathology.

## Methods

**Animals and drugs.** All studies were carried out in accordance with the National Institute of Health Guide for the Care and Use of Laboratory Animals and approved by the Vanderbilt University Institutional Animal Care and Use Committee. Mice were housed on a 12:12 light-dark cycle with lights on at 06:00. All experiments were conducted during the light phase. Food and water were available *ad libitum*. Outbred ICR mice (Harlan, Indianapolis, Indiana, USA) were used for all drug studies, mice were ordered at 5 weeks old and testing began within 2 weeks. JZL-184 (8 mg kg$^{-1}$; AbCam, Cambridge, Massachusetts, USA), Rimonabant (1 mg kg$^{-1}$; APIChem, Hangzhou, Zhejiang, China), and THC (0.25 mg kg$^{-1}$; Sigma-Aldrich, St. Louis, Missouri, USA) were prepared in dimethylsulfoxide (DMSO; Sigma-Aldrich, Milwaukee, WI, USA) and injected at a volume of 1 μl g$^{-1}$ bodyweight. DO34 (50 mg kg$^{-1}$) was synthesized as previously described[37], prepared in a 1:1:18 mixture of ethanol, kolliphor, and saline, and injected at a volume of 10 μl g$^{-1}$ bodyweight. All drugs were administered 2 h before initiation of behavioural testing.

To generate conditional DAGLα knockout mice, germline knockout mice expressing a gene-trap cassette flanked by flippase recognition target (FRT) sites were crossed with *pgk1-FLPo* mice (Tg(Pgk1-FLPo)10Sykr; Jackson Laboratories, Stock Number 011065). Following FLPo-mediated *FRT*-site recombination, the resulting conditional knockout allele consisted of *loxP*-sites flanking exon 9 of *dagla* (Fig. 6). Offspring of *pgk1-FLPo* and DAGLα$^{-/-}$ crosses were genotyped to identify founders harboring alleles that had undergone FLP-recombination (DAGLα$^{fl/+}$), and DAGLα$^{fl/+}$ founders were then bred to homozygosity (DAGLα$^{fl/fl}$). DAGLα$^{fl/fl}$ mice were maintained by homozygote x homozygote breeding. Genotypes were determined by PCR of mouse ear punch samples using the following primers (5′–3′): TGAGCCAGAGACATTTGCTG, CTGGTGAGGCCAAGTTTGTT and GGGACAGAAAACCACTTGGA. DAGLα$^{-/-}$ and DAGLα$^{fl/fl}$ mice were bred in house with behavioural testing performed with cohorts of mixed males and females that underwent stereotaxic surgery between 6 and 13 weeks of age and began behavioural testing at least 3 weeks later.

**Stereotaxic surgery.** Mice were anesthetized with isoflurane. DAGLα$^{fl/fl}$ mice (male and female 7–10 weeks old) underwent bilateral stereotaxic (Neurostar Drill and Injection Robot, Tubingen, Germany) injection of AAV5.CMV.HI.eGFP-Cre.WPRE.SV40 (AAV-GFP-CRE; titer 2 × 10$^{13}$ TU ml$^{-1}$) or AAV5.CMV.-PI.eGFP.WPRE.bGH (AAV-GFP; titer 7 × 10$^{13}$ TU ml$^{-1}$) control virus (Penn Vector Core, Philadelphia, Pennsylvania, USA). Viruses were infused into the BLA (250 nl, AP: − 1.20, ML: ± 3.35, DV: 4.95), PFC (150 nl, AP: 1.85, ML: ± 0.5, DV: 2.18), and NAc (450 nl, AP: 1.65, ML: ± 0.92, DV: 4.80) of DAGLα$^{fl/fl}$ mice at a rate of 100 nl min$^{-1}$. The syringe (10 μl Nanofil, WPI, Sarasota, Florida, USA) was first lowered (0.28 mm s$^{-1}$) to 0.3 μm deeper than the injection site, after 5 s it was raised to the injection site where it paused for 10 s before injecting. After the virus was infused, the syringe remained in place for 300 s before retracting. Surgery was counterbalanced over time and in each cage so that both conditions were represented in each cage of littermates. Behavioural testing began at least 3 weeks after viral injection.

For electrophysiological studies 3.5–5 week old ICR mice were bilaterally injected with a (2:1) mixture of AAV5.CaMKIIa.hChR2(H134R)-eYFP.WPRE.hGH (AAV-ChR2; titer 1.6 × 10$^{13}$ GC/ml) virus. The constructs were infused into the ventral hippocampus (450 nl, AP: − 2.90, ML: ± 3.25, DV: 4.16), prelimbic prefrontal cortex (110 nl, AP: 2.10, ML: ± 0.22, DV: 2.10), or lateral entorhinal cortex (350 nl, AP: 0.52 relative to lambda, ML: ± 4.56, DV: 4.12). *Ex vivo* electrophysiological recordings were performed at least 3 weeks after viral injection. All viral constructs used in this study were purchased from Penn Vector Core (Philadelphia, Pennsylvania, USA).

**Behaviour.** Foot-shock stress was performed as previously described[23]. Foot-shock stress consisted of a 7.5-min session. After a 60 s baseline, six 0.7 mA foot-shocks were delivered 1 min apart using a MED Associates fear-conditioning chamber (St. Albans, Vermont, USA). Each shock coincided with the last 2 s of a 30 s auditory tone. After an additional 60 s, mice were returned to their home cages. All post-stress behavioural testing was performed ∼24 h after completion of the final stress exposure. Behavioural responsivity to foot-shock was scored by visual assessment of behaviour from videos recorded during these foot-shock stress sessions. Scores were assigned based on the predominant behavioural response over

the 2 s shock as follows: 0-no response, 1-flinching/walking, 2-running, 3-jumping. Scores from two independent, blinded observers were averaged.

Foot-shock threshold analysis was performed at least one week after susceptibility categorization using the repeated NIH procedure described below and occurred in the same chambers as foot-shock stress. A series of 1-second foot-shocks was delivered with an interstimulus interval of 29 s with each shock increasing intensity as follows: (in mA): 0.075, 0.1, 0.15, 0.2, 0.25, 0.3, 0.35, 0.4, 0.45, 0.5, 0.55, 0.6, 0.65, 0.7. The intensity at which each mouse first flinched, ran, jumped, and vocalized was recorded and the session was terminated immediately after the first vocalization.

Repeated novelty induced hypophagia (rNIH) testing was based on previously described methods with some modifications[16,23,33]. Individually housed mice were acclimated to testing rooms under red light for at least 30 min before home-cage training and novel-cage testing. Mice were habituated to a novel, palatable food (liquid vanilla Ensure, Abbott Laboratories, Abbott Park, IL) in their home cages for 30 min per day under red light (< 50 lux) for at least 4 days before novel-cage testing. After 30 + minute acclimation in their home cages under red light illumination, mice were transferred to a novel, empty cage in a brightly lit room (∼ 300 lux; low light test in Supplementary Fig. 1 < 50 lux) and again given access to liquid vanilla Ensure for 30 min. For each mouse, the latency to drink and total weight consumed were recorded. The next week, mice underwent the same home-cage procedure for two consecutive days. After the 2nd home-cage training, mice were exposed to foot-shock stress (described above). Approximately 24 h later mice underwent novel-cage testing (novel-cage tests were performed at least 7 days apart). In drug trials, mice were injected with vehicle, JZL-184, Rimonabant, or DO34 2 h before testing. In some cohorts the same procedure was repeated 1–2 more times. Animals that did not drink in any novel-cage test were excluded from analyses.

The NIH data in Fig. 2a–f was aggregated from 3 cohorts of 40 mice that completed the baseline and stress testing as a precursor to further experiments, namely those presented in (1) Fig. 2g,h, (2) Fig. 4 and (3) Supplementary Fig. 8.

For novel open field testing (OFT), exploration of a novel open field arena contained within a sound-attenuating chamber was monitored for 10 min (27.9 × 27.9 cm; MED-OFA-510; MED Associates, St. Albans, Vermont). Beam breaks from 16 infrared beams were recorded by Activity Monitor v5.10 (MED Associates) to monitor position and behaviour.

Light-dark box testing (LD) was performed as previously described[16,27]. Exploration of open field chambers containing dark box inserts that split the chamber into light (∼ 300 lux) and dark (< 5 lux) halves (ENV-511; MED Associates, St. Albans, Vermont) was recorded by Activity Monitor v5.10 as above. Position and behaviour were monitored as described above for 10 min.

The elevated plus maze (EPM) consisted of two open arms (30 × 10 cm) and two closed arms (30 × 10 × 20 cm) that met at a centre junction (5 × 5 cm). The apparatus was elevated 50 cm from the floor. Light levels in the open arms were approximately 200 lux, while the closed arms were < 100 lux. Mice were placed in the centre of the maze, facing a closed arm, and allowed to explore for 5 min. ANY-maze (Stoelting, Wood Dale, Illinois, USA) video-tracking software was used to monitor and analyse behaviour during the test.

**LC/MS/MS detection of lipids.** Mice underwent cervical dislocation immediately followed by decapitation. The brain was quickly removed, placed in a brain matrix, and covered with ice cold NMDG-ACSF (details in electrophysiology section below). 1–2 mm thick coronal sections containing the target brain regions were frozen on a metal block in dry ice. Dissections were performed on the frozen tissue for the production of amygdala-, nucleus accumbens-, PFC-, and ventral hippocampus-enriched samples using a 1 or 2 mm diameter metal micropunch. Samples were stored at − 80 °C until extraction.

LC/MS/MS detection of endocannabinoids and arachidonic acid was performed as previously described with minor modifications[16]. Briefly, all samples were homogenized directly in methanol, incubated at − 20 °C overnight, and centrifuged at 10 g for 15 min at 4 °C; water was added to the supernatant for a final ratio of 70:30 Methanol:Water. Sample (20 μl) was injected into a C-18 column (50 × 2 mm, 3 μm; Phenomenex or 50 × 2.1 mm, 1.7 μm; Acquity) under either of the following two conditions:

(1) 20% A (water with 80 μM silver acetate and 0.1% glacial acetic acid (v/v)) and 80% B (methanol with 80 μM silver acetate and 0.1% glacial acetic acid (v/v)) from 0 to 0.5 min, increased to 0% A and 100% B from 0.5 to 3.5 min and held for 1 min, and returned to 20% A and 80% B from 4.5 to 6.5. Analytes were detected via selective reaction monitoring (as $[M + Ag] +$ complexes except AA, which is ionized as $[(M–H) + 2Ag] +$) in the positive ion mode using the following reactions (the mass in parentheses represents the mass of the deuterated internal standard): AA ($m/z$ 519(527)→409(417)); 2-AG ($m/z$ 485(493)→411(419)); and AEA ($m/z$ 454(462)→432(440)) using a Sciex QTrap 6500 mass spectrometer.

(2) 65% A (water with 0.1% formic acid (v/v)) and 35% B (2:1 acetonitrile:-methanol with 0.1% formic acid (v/v)) from 0 to 0.15 min, increased to 1% A and 99% B from 0.15 to 5 min and held for 1.8 min, and returned to 65% A and 35% B from 6.8–7.2. Analytes were detected via selective reaction monitoring in either the positive ion mode (2-AG and AEA as $[MH] +$ complexes) or

negative ion mode (AA as [M-H]-) using the following reactions (the mass in parentheses represents the mass of the deuterated internal standard): 2-AG ($m/z$ 379(384)→287(287)); AEA ($m/z$ 348(352)→62 (66)); and AA ($m/z$ 303(311)→259(267)) using a Sciex QTrap 6500 mass spectrometer.

Quantification was achieved via stable-isotope dilution for AA, 2-AG and AEA.

**Corticosterone ELISA.** Trunk blood was collected in tubes containing 25 µl EDTA (128 µM) immediately following cervical dislocation and decapitation. Gentle inversion of tubes to promote mixing of the blood and the EDTA prevented coagulation. Samples were kept on ice. Plasma was then extracted via centrifugation and stored at $-80\,°C$. A corticosterone ELISA assay was performed on the extracted plasma using a commercial kit (Enzo Life Science, Catalog No. ADI-900-097). The ELISA plate was read at 405 nm. The online tool MyAssays (Cayman Chemicals) was used to calculate the corticosterone concentration of each sample.

**Western blot.** Tissue was dounce homogenized in lysis buffer containing 50 mM HEPES (pH 7.5), 0.5 mM TCEP, 10 µg ml$^{-1}$ leupeptin, 10 µg ml$^{-1}$ pepstatin, 1 mM PMSF, 2 mM EDTA and PhosStop tablets (Roche). Total protein was measured in lysates by Bradford protein assay, and equal protein was loaded and resolved by SDS-PAGE and transferred to nitrocellulose membranes for western blotting using rabbit anti-DGLa (1:6,000), rabbit anti-CB1R (1:1,000), and rabbit anti-MGL (1:750)[69]. Densitometry was performed using Image J (National Institutes of Health, Bethesda, MD), and signals were normalized for protein loading by dividing the individual band area by Ponceau S staining. Full, uncropped blots are shown in Supplementary Fig. 13.

**Immunohistochemistry.** Following stereotaxic injections of AAV-GFP-Cre or AAV-GFP and behavioural analyses, mice were anesthetized with isoflurane and transcardially perfused with 10 ml of phosphate buffered saline (PBS, pH 7.4), followed immediately by 15 ml of 4% paraformaldehyde (PFA). Brains were dissected and post-fixed for 24 h in 4% PFA, and then transferred to 30% sucrose solution for 4–5 days. Brains were cut at 40 µm using a Leica 3050-S cryostat (Leica Biosystems, Buffalo Grove, IL, USA). Slices were rinsed 3X for ten minutes in tris-buffered saline (TBS, pH 7.4) and subsequently washed for 30 min in TBS + (4% horse serum, and 0.2% Triton X-100). The slices were then incubated overnight at room temperature in TBS + solution containing rabbit anti-DAGLα primary antibody[69,70] at 1:500 concentration. Slices were rinsed 3X for ten minutes in TBS + and then incubated in TBS + containing Alexa Fluor 546 donkey anti-rabbit secondary antibody (ThermoFisher Scientific, Waltham, MA, USA) at 1:1000 concentration for 2.5 h. Slices were then rinsed 3X for ten minutes in TBS before being mounted onto slides with 0.15% gelatin solution. Slides were cover-slipped with DPX mountant, and imaged on an upright Zeiss Axio Imager M2 microscope (Zeiss, Oberkochen, Germany).

***Ex vivo* electrophysiological recordings.** Mice used for electrophysiology were drug-naïve, but where noted, mice were foot-shock stressed 1 day before sacrifice for electrophysiological recordings. Mice were briefly anesthetized with isoflurane and transcardially perfused with ice-cold oxygenated (95% v/v O$_2$, 5% v/v CO$_2$) N-methyl-D-glucamine (NMDG) based ACSF[71] comprised of (in mM): 93 NMDG, 2.5 KCl, 1.2 NaH$_2$PO$_4$, 30 NaHCO$_3$, 20 HEPES, 25 glucose, 5 Na-ascorbate, 3 Na-pyruvate, 5 N-acetylcystine, 0.5 CaCl$_2 \cdot 4$H$_2$O and 10 MgSO$_4 \cdot 7$H$_2$O. The brain was quickly removed and 250 µm coronal slices of the basolateral amygdala (BLA) were cut using a Leica VT1000S vibratome (Leica Microsystems, Bannockburn, IL, USA) in the NMDG solution. Slices were incubated for 8–10 min at 32 °C oxygenated NMDG-ACSF and stored at 24 °C until recording in HEPES-based ACSF containing (in mM): 92 NaCl, 2.5 KCl, 1.2 NaH$_2$PO$_4$, 30 NaHCO$_3$, 20 HEPES, 25 glucose, 5 ascorbate, 3 Na-pyruvate, 5 N-acetylcystine, 2 CaCl$_2 \cdot 4$H$_2$O and 2 MgSO$_4 \cdot 7$H$_2$O. Slices from each animal were incubated for 2–3 h in vehicle and Rimonabant (5 µM) or JZL-184 (1 µM) containing ACSF. The order of recording from each condition was alternated day by day to control for slice age and incubation time. Recordings were performed in a submerged recording chamber during continuous perfusion of oxygenated ACSF containing (in mM): 113 NaCl, 2.5 KCl, 1.2 MgSO$_4 \cdot 7$H$_2$O, 2.5 CaCl$_2 \cdot 2$H$_2$O, 1 NaH$_2$PO$_4$, 26 NaHCO$_3$, 1 ascorbate, 3 Na-pyruvate and 20 glucose; at a flow rate of 2.5–3 ml min$^{-1}$. For Supplementary Fig. 12, the CB1R agonist CP-55940 (5 µM) was washed onto slices following acquisition of a stable baseline. For all experiments, and for drug solutions, 0.5 g l$^{-1}$ of fatty acid free bovine serum albumin (Sigma-Aldrich, St. Louis, MO, USA) was also added to the ACSF to increase solubility of the lipophilic drugs, and to minimize nonspecific binding of these compounds.

Slices were visualized using a Nikon microscope equipped with differential interference contrast video microscopy. Whole-cell voltage-clamp recordings from BLA pyramidal cells held at $-70$ mV were obtained under visual control using a 40x objective. 2–3 MΩ borosilicate glass pipettes were filled with high [K$^+$] based solution containing (in mM): 125 K$^+$-gluconate, 4 NaCl, 10 HEPES, 4 Mg-ATP, 0.3 Na-GTP, and 10 Na-phosphocreatine. Only cells with access resistance $<20$ MΩ were included. For spontaneous EPSC (sEPSC) measurements ACSF was supplemented with the GABA$_A$ receptor antagonist picrotoxin (50 µM; Abcam, Cambridge, MA) to pharmacologically isolate glutamatergic transmission[16,27,72],

for optically induced EPSC (oEPSC) recordings picrotoxin (50 µM) was applied in the internal solution to avoid optically induced population activity. For optical stimulation 1–2 ms 480 nm blue light pulse was delivered by an LED (ThorLabs) directed through the objective. Intensity curves indicate current to the LED in amperes. Optically evoked depolarization-induced suppression of excitation (oDSE) was examined under voltage-clamp conditions where cells were recorded at a holding potential of $-70$ mV. Excitatory postsynaptic currents (EPSCs) were elicited at a rate of 0.2 Hz. To induce DSE, a depolarizing pulse ($-70$ to $+30$ mV) was applied to the postsynaptic neuron for 10 s. Within each DSE trial, EPSC amplitudes were normalized to the averaged baseline response, and maximum DSE was classified as the first EPSC following the depolarizing pulse. Recordings were performed using a MultiClamp 700B amplifier (Molecular Devices), and Clampex software (version 10.2; Molecular Devices).

**Statistical analysis.** All statistical analyses were performed with GraphPad Prism 6 (San Diego, CA, USA). Student's $t$-test, one-way ANOVA, and two-way ANOVA were used as appropriate. One- and two-way ANOVA were followed by post-hoc Sidak/Holm-Sidak multiple comparisons tests. Relevant F and $P$ values for one- and two-way ANOVA are shown in individual figure panels. $P$ values (and/or asterisks denoting significance as follows: *$P<0.05$, **$P<0.01$, ***$P<0.001$, and ****$P<0.0001$) for pairwise comparisons derived from post-hoc testing or from $t$-tests are shown in individual figure panels, with details in figure legends. $R^2$ and $P$ values for linear regression analyses are shown in all correlation panels. Chi-squared analyses were performed to compare susceptibility ratios and the resultant $P$ values are reported in each figure panel. Rout test for outlier identification was used. Testing was counterbalanced, but no randomization was performed, and sample sizes were derived empirically during the course of the experiments guided by our previous work using these assays[16,23,33]. Experimenters were blinded to treatment condition during experimentation. Data are presented as mean ± s.e.m.

**Data availability.** The data herein are available from the corresponding author (S.P.) upon reasonable request.

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

## Acknowledgements

This work was supported by NIH grants F31MH106192 (R.J.B.), R01NS078291 (R.J.C.), R01MH100096, R01MH107435, and NARSAD Independent Investigator (S.P.) and Young Investigator Awards (R.B.) from the Brain & Behavior Research Foundation. Part of the research in this publication was also supported by the Hobbs Foundation Award. Behavioral studies were carried out at the Vanderbilt Neurobehavioral Core (supported by the EKS NICHD Award U54HD083211) and mass spectrometry studies were carried out at the Vanderbilt University Mass Spectrometry Research Center. The content of the work is solely the responsibility of the authors and does not necessarily represent the official views of the NIH.

## Author contributions

R.J.B. conducted behavioural and neurochemical experiments with assistance from A.H., N.D.H., A.D.G., R.M.-B., J.B. and D.M. in laboratory of S.P. R.B. conducted electrophysiological studies in laboratory of S.P. B.C.S. assisted in generation of DAGLα$^{f/f}$ mice in laboratory of R.J.C. B.C.S. and W.P.P. completed biochemical experiments in the laboratory of R.J.C., K.M., L.M., M.J.U., D.G.W. and R.J.C. contributed to generation of reagents. R.J.B. and R.B. contributed to research design, data interpretation and analysis in laboratory of S.P. R.J.B. and S.P. wrote the manuscript with input from all authors.

## Additional information

**Competing interests:** S.P. and L.M. have an active collaborative research contract with Lundbeck Pharmaceuticals, however, the work presented in the paper was supported solely by the NIH, the Brain & Behavior Research Foundation, and Vanderbilt University.

