## [Peer Review File · Nature Communications]

Reviewers' comments:

Reviewer #1 (Remarks to the Author):

In the current manuscript Bluett and colleagues address the important topic of stress resilience. To study this phenomenon, they utilize acute or repeated foot shock stress exposure. Animals are subsequently subdivided in susceptible and resilient based on their post-stress induced anxiety level in the NIH test. Consequently, the authors provide compelling evidence that 2-AG - CB1 signaling promotes stress resilience, i.e. a lack of stress-induced anxiety, while blocking this pathway enhances stress susceptibility. Finally, the authors started to dissect the local circuitry and provided initial data implicating the BLA and vHip-BLA glutamatergic projections in this phenomenon. Strengths of the paper include a clear rationale, a logical and conclusive set of experiments and a clear and transparent representation of the data, including the statistical analysis. On the other hand I noted a number of shortcomings that I would like the authors to address:

- a. In the initial experiments the authors confirmed the anxiety phenotype using additional anxiety tests (EPM, OFT, DaLi) to support the general nature of their conclusions and to exclude specific effects of 2-AG manipulations on appetite and food intake. However, this important control is missing in the decisive experiments using repeated stress exposure with/without pharmacological treatment (figures 4/5). I believe this important control should be added to fully exclude this possibility.
- b. Once the authors start dissecting the 2-AG effects in the amygdala, they extend their stress protocol to 5 days, as no effects were seen 1 day after stress. They now throw in the concept of "stress adaptation" and somehow equal this to the concept of resilience. I think this is a mistake and the two phenomena should be viewed separately. Adaptation to a homotypic stressor is clearly different to stress resilience, which is viewed as the response to unpredictable (acute or chronic) stressors. While it is interesting that the authors observe effects on stress adaptation when manipulating 2-AG in the amygdala, this should not be mixed with conclusions on stress resilience.
- c. Did the authors consider using conditional CB1 receptor deletion, e.g. in the vHip?
- d. As the initial data were mainly based on pharmacological manipulation of the endogenous cannabinoid system, the same strategy should also be followed for the region-specific interventions, i.e. local administration of JZL or DO34.
- e. How long-lasting are the effects of an acute foot shock stress on anxiety levels?
- f. In Figure 7C the number of susceptible animals is given as 10, while everywhere else it is given as 11. Please check and clarify.
- g. Check the formatting of the reference and correct the multiple errors!

Reviewer #2 (Remarks to the Author):

Nature Communications Review of "An Endocannabinoid Mechanism Promoting Resilience to Traumatic Stress"

Summary

This is an interesting, novel, well-written, and important manuscript providing further mechanistic insight into the role of endogenous cannabinoids in modulating stress, in particular, basolateral amygdala (BLA) neural circuits. With targeted pharmacological, inducible knockouts, optogenetic and behavioral approaches, the authors elegantly show that DAGL/2-AG are necessary and sufficient for a resilient phenotype following repeated stress exposure. Furthermore, there data suggest that 2-AG may be important in the BLA for the ventral hippocampal to BLA projection mediating anxiety-like behavior and novelty induced hypophagia. My enthusiasm is only somewhat limited by the below remaining concerns.

Concerns

The authors show that their manipulations demonstrate both increased 2-AG and decreased AA signaling. How do they know that the decreased AA signaling are not accounting for some of the behavioral effects?

In figure 6, The DAGLa IHC figure looks a little strange, with all or no signal, it would be nice to see a bit lower power so that one could appreciate the borders of the virus injection, as well as considering in situs of DAGLa after AAV-Cre injection as another way to convincingly determine targeted DAGLa knockout.

In suppl Figure 6, the AAV-Cre group has a ratio of 8:7 resilient:susceptible and the AAV-GFP control has 5:12 resilient vs. Susceptible. This raises the question of whether the nonsignificant effects that are primarily referred to in results and discussion are actually driven by low power (fewer animals tested in the PFC study than the BLA study) thus giving a false impression that DAGLa may not be needed in PFC for resilience, when in fact it may.

My reading of the methods states that mice were ordered at 5 weeks, and testing began within 2 weeks - so these could be considered adolescent / late adolescent age mice, do you think you would see similar effects in fully-grown adults? This would seem to at least deserve comment in the discussion.

Regarding the DAGL inhibitor studies with DO34 - these results raise the question as to whether different endocannabinoids lead to different behaviors Thus I wonder if it is a bit overstated that the MAGL inhibition - and 2AG increase - results in resilience and DAGL inhibition and AEA have no effect. Rather, I wonder if the effects are more subtle and different, such that the different eCBs may modulate different types of resilience (or otherwise) behaviors?

Minor:

Abstract- second or third paragraph would be good to mention that this study is in rodents. Further down it is stated that the data are in mice, but could be confusing given the wording in the initial abstract statements.

The second paragraph reads like a listing of prior data about eCBs but is not particularly

well organized or focused and the writing could be improved to set up the current findings and story.

In a few places the authors refer to 'anxiety' or 'generalized anxiety' in the mice - I would suggested 'anxiety-like behaviors'.

Reviewer #3 (Remarks to the Author):

The work by Bluett et al. investigated individual differences in endocannabinoid system alterations after traumatic stress. It was found that such alterations occurred in the BLA and that augmenting 2-AG signaling by pharmacological means could convert susceptible mice to resilient mice.

This study is very extensive and highly interesting. Notably, a new behavioral paradigm was established in order to investigate stress resilience in mice. Using this new paradigm, in combination with extensive behavioral analysis, pharmacological and genetic interventions, and in vitro electrophysiology, the authors were able to significantly advance the insights of endocannabinoid functions in stress response and in stress resilience.

Major points:

(1) Although e.g., anxiety-like behavior was tested prior to shock, it must also be tested whether there are individual differences in pain sensitivity, behavioral response during foot shock, behavioral habituation to shock (as several shocks were delivered), and/or in plasma cortisol levels immediately after shock delivery. Such parameters could explain the differences seen later in feeding latency. Thus, individual differences (e.g., in sensory perception) prior to the traumatic event of the shock would influence the impact of the shock on the neurobiological consequences of the shock, such as alterations in neural excitability and neurotransmission/synaptic plasticity processes.

(2) What are the long-term effects of the shocks? See Krishnan et al. (2007), where mice were tested several weeks after chronic social defeat. The long-term effects of the traumatic event are fundamental for making conclusions on resilience versus susceptibility, as otherwise rather short-term / acute effects would have been monitored, and this would not go much beyond research on acute stress responses.

(3) JZL184 (8 mg/kg) decreased feeding latency in particular in the case when animals were stressed (Figure 1d). Later in the work, it becomes evident that oDSE is reduced in susceptible mice as compared to resilient mice (Figure 8). In the light that cannabinoid action on feeding is bimodal (see Bellocchio et al. 2010), one would expect that dose-dependent effects of JZL184 in stressed versus non-stressed on feeding latency are present.

(4) It is important to perform local JZL184 injections into the BLA. This experiment is essential in order to validate the proposed mechanism. The BLA-specific deletion of DAGL-alpha is not sufficient, as this intervention led to increased vulnerability.

(5) In order to substantiate the proposed mechanism, measurements of eCBs in BLA, PFC, and vHIP in resilient versus vulnerable mice are required. Furthermore, as a dysregulation

of eCB system activity is proposed to be present, it will be interesting to monitor DAGL-alpha, CB1, and MAGL protein levels in the BLA, PFC, and vHIP. The latter two regions may serve as controls where no alterations may be expected. It might also be the case that CB1 mRNA levels are decreased in the hippocampus, and thereby influencing DSE in the BLA. Generally: Are there differences in basal and/or behaviorally induced states between resilient and vulnerable mice in these parameters mentioned above?

(6) Fig. 2a: Scheme is not clear. Days 6-9 are missing. The presentation implies a temporal progression, but it rather depicts there different blocks in the experimental set-up. The abbreviation FS-NC is not explained. Please improve this scheme.

(7) Figure 8a-c: The authors should present the relevant comparison between Veh Res versus Veh Susc. Are there significant differences?

(8) Figure 8, Suppl Figure 8: Although oDSE in vHIP-BLA projection is clearly altered in resilient versus susceptible mice (Figure 8), and although PreL-BLA oDSE is less strong than vHIP-BLA oDSE, it cannot be excluded that PreL-BLA oDSE is also different between resilient versus susceptible mice. Thus, this study did not formally prove that the 2-AG-CB1 signaling of the vHIP-BLA projection is responsible for the behavioral difference between resilient and susceptible mice. Only an association of altered oDSE with susceptibility was shown.

(9) Are there differences in DSE in BLA without optic stimulation between resilient and susceptible mice?

(10) The authors must mention and discuss some work on stress/traumatic stress - amygdala function - cannabinoids, performed by the group of Irit Akirav. This work fits to the present one, where THC alleviated the stress effects.

(11) The authors may wish to see this work in the context of other investigations on stress resilience and alterations in synaptic transmission / excitability, such as done by Bo Li and Eric Nestler.

Minor points:

(1) Please term the behavioral outcome as "suppression of feeding" and do not term this behavior from time to time as anxiety behavior.

(2) Page 23: Please specify type of LC-MS instrument used. Unfortunately, the authors did not use the current state-of-the-art method for endocannabinoid / lipid quantifications, which is MS/MS, whereby the identity of the ions can be clearly uncovered.

(3) Figure 1, and all other figures: Please explain abbreviations always in the legend. In Figure 1: NIH.

(4) Figure 1a-c: Explicitly mention the group size, as the individual points are not visible, as in particular the blue bars covers the blue dots.

(5) Figure 1d,e, Figure 4a,b,e,f: The treatment with rimonabant is not very meaningful, as it is completely dominant in the influence on feeding latency and food consumption. Please comment. In Fig. 1, RIM was given at a dose of 1 mg/kg. Was the same dose given in the experiments shown in Figure 4?

(6) Figures 3, 4 etc.: Please indicate in the graphs the meaning of the x-axis (days). Also indicate by arrows the time points of shock delivery.

(7) Figure 4d: Please add "Feeding" to the y-axis.

(8) Figure 4i: While for the resilient group the color is kept in red, for the logic for the black, blue and orange colors are not clear.

(9) Figure 6c: The immunostaining for DAGL-alpha is of insufficient quality. This protein should show a clear postsynaptic staining, which is not visible in the "GFP" group.

We would like to thank the reviewers for their unanimously positive appraisal of our study. We have made every effort to address the concerns of the reviewers as noted point-by-point below. In the vast majority of cases we have added substantial new data to improve the quality of the study and the validity of the conclusions (listed briefly below). There are some instances where we were unable to completely address reviewer critiques, however, we feel that overall the paper provides substantial and high-quality evidence for the conclusions, presents highly novel findings, and has the potential to impact the field in a meaningful way. We hope the reviewers will agree that the work as presented represents a significant conceptual advance in the field and that publication and dissemination of our result is now justified.

Key additional new data now provided:

- 1) Time course of the anxiety-like effect of acute foot-shock stress 3d and 14d after stress (Supplementary Fig. 3)
- 2) Measurement of 2-AG tissue levels and CB1, DAGL α , MAGL protein levels in amygdala, PFC, nucleus accumbens, and vHIP in stress susceptible and resilient mice (Supplementary Fig. 6-7)
- 3) Additional control studies indicating a lack of correlation between feeding latency and food consumption in the repeated stress NIH assay confirming the effects of JZL and DO34 were not driven by changes in appetite (Fig. 4k and 5j)
- 4) Additional controls showing DO34 increases anxiety in the elevated-plus maze in the repeated stress paradigm further confirming the effects of 2-AG modulation are not caused by changes in appetite (Fig. 5k).
- 5) Evaluation of differences in shock responsivity, conditioned freezing, shock habituation, and corticosterone levels in stress susceptible and resilient mice (Supplementary Fig. 2c-h).
- 6) Additional optogenetic electrophysiological studies showing no differences at PFC-BLA synapses in either synaptic connectivity or eCB signaling, providing specificity for effects observed at vHIP-BLA synapses (Fig. 9h-m).
- 7) Evaluation of functional CB1 signaling at vHIP-BLA synapses between stress susceptible and resilient mice using electrophysiological approaches (Supplementary Fig. 12).
- 8) Increased sample size for PFC and nucleus accumbens-specific DAGL α deletion behavioral experiments (Supplementary Fig. 10).

Reviewers' comments:

Reviewer #1 (Remarks to the Author):

In the current manuscript Bluett and colleagues address the important topic of stress resilience. To study this phenomenon, they utilize acute or repeated foot shock stress exposure. Animals are subsequently subdivided in susceptible and resilient based on their post-stress induced anxiety level in the NIH test. Consequently, the authors provide compelling evidence that 2-AG - CB1 signaling promotes stress resilience, i.e. a lack of stress-induced anxiety, while blocking this pathway enhances stress susceptibility. Finally, the authors started to dissect the local circuitry and provided initial data implicating the BLA and vHip-BLA glutamatergic projections in this phenomenon. Strengths of the paper include a clear rationale, a logical and conclusive set of experiments and a clear and transparent representation of the data, including the statistical analysis. On the other hand I noted a number of shortcomings that I would like the authors to address:

a. In the initial experiments the authors confirmed the anxiety phenotype using additional anxiety tests (EPM, OFT, DaLi) to support the general nature of their conclusions and to exclude specific effects of 2-AG manipulations on appetite and food intake. However, this important control is missing in the decisive experiments using repeated stress exposure with/without pharmacological treatment (figures 4/5). I believe this important control should be added to fully exclude this possibility.

We have added an experiment demonstrating that DO34 increases anxiety, relative to vehicle treatment, in the elevated-plus maze in the context of the repeated foot-shock paradigm, reducing the possibility that its effects in NIH after multiple stress exposures are primarily driven by consummatory changes (Fig. 5k). More importantly, we now show a clear lack of correlation between novel-cage consumption and feeding latency after vehicle treatment, JZL184 treatment and DO34 treatment; if changes in appetite are driving changes in feeding latency there should be a correlation between these measures, which there is not (Fig. 4k and 5j). In other words, without an association between the two measures there can be no causal relationship. We have previously shown that the consummatory and latency aspects of this assay are dissociable (Gamble-George et al 2013), which is consistent with these

correlational analyses. We were not able to detect an anxiolytic like effect of JZL184 in the EPM (Supplementary Fig. 4), however this may be due to the reduced sensitivity of this assay to 2-AG augmentation. Indeed, a recent study found that MAGL inhibition was not effective in reducing anxiety in the EPM in high anxiety mice (Morena et al 2016). However, Figure 1L clearly shows anxiolytic effects of JZL184 in the light dark box after single stress exposure.

b. Once the authors start dissecting the 2-AG effects in the amygdala, they extend their stress protocol to 5 days, as no effects were seen 1 day after stress. They now throw in the concept of "stress adaptation" and somehow equal this to the concept of resilience. I think this is a mistake and the two phenomena should be viewed separately. Adaptation to a homotypic stressor is clearly different to stress resilience, which is viewed as the response to unpredictable (acute or chronic) stressors. While it is interesting that the authors observe effects on stress adaptation when manipulating 2-AG in the amygdala, this should not be mixed with conclusions on stress resilience. We agree that it is important to be very clear about these different aspects of stress responsivity and have endeavored to make our results and discussion of these concepts more explicit with regard to this distinction.

c. Did the authors consider using conditional CB1 receptor deletion, e.g. in the vHip?

We do not currently have this mouse line, but are trying to obtain it to do further experiments using this model in the future.

d. As the initial data were mainly based on pharmacological manipulation of the endogenous cannabinoid system, the same strategy should also be followed for the region-specific interventions, i.e. local administration of JZL or DO34.

Unfortunately, this experiment is outside our current technical expertise, and represents a substantial investment in training and time that was not feasible in the 90-day timeframe for a revision. We did generate a DAGL α overexpression AAV, however after injection of 80 mice and behavioral analysis the results were inconclusive likely due to low viral expression; we were not able to validate overexpression in the limited time we had and optimization of this tool could take some time. Please note we have anatomical specificity for our effects with region-specific DAGL deletions, and that in combination with the totality of other converging data supporting the conclusions, feel the paper is still of high quality and broad interest.

e. How long-lasting are the effects of an acute foot shock stress on anxiety levels?

We have now shown that susceptible individuals are still more anxious than resilient individuals at 3 days and at 2 weeks after foot-shock stress, although the magnitude of the effect clearly trends down over the 2 week period (Supplementary Fig. 3).

f. In Figure 7C the number of susceptible animals is given as 10, while everywhere else it is given as 11. Please check and clarify.

Thank you, we have fixed this and associated statistics.

g. Check the formatting of the reference and correct the multiple errors!

They have been fixed.

Reviewer #2 (Remarks to the Author):

This is an interesting, novel, well-written, and important manuscript providing further mechanistic insight into the role of endogenous cannabinoids in modulating stress, in particular, basolateral amygdala (BLA) neural circuits. With targeted pharmacological, inducible knockouts, optogenetic and behavioral approaches, the authors elegantly show that DAGL/2-AG are necessary and sufficient for a resilient phenotype following repeated stress exposure. Furthermore, there data suggest that 2-AG may be important in the BLA for the ventral hippocampal to BLA projection mediating anxiety-like behavior and novelty induced hypophagia. My enthusiasm is only somewhat limited by the below remaining concerns.

The authors show that their manipulations demonstrate both increased 2-AG and decreased AA signaling. How do they know that the decreased AA signaling are not accounting for some of the behavioral effects?

Both JZL-184 and DO34 treatments reduce AA in the amygdala and yet have opposite effects on stress-induced

anxiety, making it extremely unlikely that alterations in AA-related signaling mediate our behavioral effects. This new mass spectrometry data is shown in Figures 1b and 5b.

In figure 6, The DAGLa IHC figure looks a little strange, with all or no signal, it would be nice to see a bit lower power so that one could appreciate the borders of the virus injection, as well as considering in situ of DAGLa after AAV-Cre injection as another way to convincingly determine targeted DAGLa knockout.

Alternate images of the DAGLa IHC in Figure 6 have been provided to better compare DAGL expression in AAV-GFP and AAV-GFP-CRE injected slices. Please note the main form of validation comes from the 2-AG measurement from a microdissection of the injection site itself which is also shown in Figure 6.

In suppl Figure 6, the AAV-Cre group has a ratio of 8:7 resilient:susceptible and the AAV-GFP control has 5:12 resilient vs. Susceptible. This raises the question of whether the nonsignificant effects that are primarily referred to in results and discussion are actually driven by low power (fewer animals tested in the PFC study than the BLA study) thus giving a false impression that DAGLa may not be needed in PFC for resilience, when in fact it may. AAV-GFP and AAV-GFP-CRE cohort sizes have been increased for the PFC and NAC groups to ~20 per brain region. The conclusions have not changed based on the increase in sample size. These data can now be found in Supplementary Fig. 10.

My reading of the methods states that mice were ordered at 5 weeks, and testing began within 2 weeks - so these could be considered adolescent / late adolescent age mice, do you think you would see similar effects in fully-grown adults? This would seem to at least deserve comment in the discussion.

We have mentioned this caveat in the discussion – given that several of the cohorts underwent 3-4 weeks of testing (at which point they would be considered adults) and all of the virally-injected mice were adults by the time they were tested, we do believe that we would see similar results, although we cannot definitively say that since it is possible that the adolescent social isolation may modulate stress responses.

Regarding the DAGL inhibitor studies with DO34 - these results raise the question as to whether different endocannabinoids lead to different behaviors. Thus I wonder if it is a bit overstated that the MAGL inhibition - and 2AG increase - results in resilience and DAGL inhibition and AEA have no effect. Rather, I wonder if the effects are more subtle and different, such that the different eCBs may modulate different types of resilience (or otherwise) behaviors?

AEA certainly plays a complex and interesting role in regulating stress-related processes and it is possible that AEA plays a role in resilience; we did not intend to imply that it does not, simply that we have not tested it here and that the manipulations herein do not affect AEA levels making it most likely that these particular effects are dependent on 2-AG signaling. Clearly subsequent studies using this paradigm could be conducted with FAAH inhibitors and COX-2 inhibitors to test the role of other eCBs and metabolic enzyme targets; which we are planning to do in the near future.

Minor:

Abstract- second or third paragraph would be good to mention that this study is in rodents. Further down it is stated that the data are in mice, but could be confusing given the wording in the initial abstract statements.

We have added this to the abstract.

The second paragraph reads like a listing of prior data about eCBs but is not particularly well organized or focused and the writing could be improved to set up the current findings and story.

We have changed the second paragraph to be shorter and more concise in terms of introducing the relevant eCB ligands and enzymes which we feel are important background for the reader. We then provide more “set-up” for the current study in paragraph 3.

In a few places the authors refer to 'anxiety' or 'generalized anxiety' in the mice - I would suggest 'anxiety-like behaviors'.

We have changed to 'anxiety-like behavior'.

Reviewer #3 (Remarks to the Author):

Major points:

The work by Bluett et al. investigated individual differences in endocannabinoid system alterations after traumatic stress. It was found that such alterations occurred in the BLA and that augmenting 2-AG signaling by pharmacological means could convert susceptible mice to resilient mice.

This study is very extensive and highly interesting. Notably, a new behavioral paradigm was established in order to investigate stress resilience in mice. Using this new paradigm, in combination with extensive behavioral analysis, pharmacological and genetic interventions, and in vitro electrophysiology, the authors were able to significantly advance the insights of endocannabinoid functions in stress response and in stress resilience.

(1) Although e.g., anxiety-like behavior was tested prior to shock, it must also be tested whether there are individual differences in pain sensitivity, behavioral response during foot shock, behavioral habituation to shock (as several shocks were delivered), and/or in plasma cortisol levels immediately after shock delivery. Such parameters could explain the differences seen later in feeding latency. Thus, individual differences (e.g., in sensory perception) prior to the traumatic event of the shock would influence the impact of the shock on the neurobiological consequences of the shock, such as alterations in neural excitability and neurotransmission/synaptic plasticity processes.

In order to address this point we have added experiments demonstrating that resilient and susceptible individuals do not differ in the foot-shock thresholds that elicit various behavioral responses, in freezing to the conditioned stimulus across two stress sessions, or in stress-induced increases in plasma corticosterone (Supplementary Fig. 2c-h).

(2) What are the long-term effects of the shocks? See Krishnan et al. (2007), where mice were tested several weeks after chronic social defeat. The long-term effects of the traumatic event are fundamental for making conclusions on resilience versus susceptibility, as otherwise rather short-term / acute effects would have been monitored, and this would not go much beyond research on acute stress responses.

We have now shown that susceptible individuals are still more anxious than resilient individuals at 72 hours after foot-shock stress and at 2 weeks after foot-shock stress, although the magnitude of the effect clearly trends down over the 2 week period (Supplementary Fig. 3).

(3) JZL184 (8 mg/kg) decreased feeding latency in particular in the case when animals were stressed (Figure 1d). Later in the work, it becomes evident that oDSE is reduced in susceptible mice as compared to resilient mice (Figure 8). In the light that cannabinoid action on feeding is bimodal (see Bellocchio et al. 2010), one would expect that dose-dependent effects of JZL184 in stressed versus non-stressed on feeding latency are present.

It is possible that the effects are bimodal, but we have conducted dose-response studies in this assay for other projects and do not find this to be the case (see figure below with NIH feeding latency data from 5, 10, and 15 mg/kg JZL-184 treatment compared to vehicle treatment). Moreover, this argues that the effects on latency are not driven by cannabinoid effects on appetite as summarized in the response to reviewer 1a above. Additionally, at doses above those shown below, JZL-184 can produce locomotor and other side effects, the presence of which would complicate interpretation of both latency and consumption measures.

(4) It is important to perform local JZL184 injections into the BLA. This experiment is essential in order to validate the proposed mechanism. The BLA-specific deletion of DAGL-alpha is not sufficient, as this intervention led to increased vulnerability.

Unfortunately, this experiment is outside our current technical expertise, and represents a substantial investment in

training and time that was not feasible in the 90-day timeframe for a revision. We did generate a DAGL α overexpression AAV, however after injection of 80 mice and behavioral analysis the results were inconclusive likely due to low viral expression; we were not able to validate overexpression in the limited time we had and optimization of this tool could take some time. Please note we have anatomical specificity for our effects with region-specific DAGL deletions, and that in combination with the totality of other converging data supporting the conclusions, feel the paper is still of high quality and broad interest.

(5) In order to substantiate the proposed mechanism, measurements of eCBs in BLA, PFC, and vHIP in resilient versus vulnerable mice are required. Furthermore, as a dysregulation of eCB system activity is proposed to be present, it will be interesting to monitor DAGL-alpha, CB1, and MAGL protein levels in the BLA, PFC, and vHIP. The latter two regions may serve as controls where no alterations may be expected. It might also be the case that CB1 mRNA levels are decreased in the hippocampus, and thereby influencing DSE in the BLA. Generally: Are there differences in basal and/or behaviorally induced states between resilient and vulnerable mice in these parameters mentioned above?

We have added regional 2-AG measurements for all brain regions (Supplementary Fig. 7). While overall tissue levels of 2-AG do not correlate with stress resilience, Larry Parsons' work using microdialysis has suggested the likelihood that overall tissue levels are not necessarily representative of signaling capacity – as supported by our electrophysiology measurements. We have also included Western blot data showing no significant differences in protein levels of 2-AG metabolizing enzymes between stress susceptible and resilient mice (Supplementary Fig. 6). Regarding the hippocampal CB1 mRNA, we opted to rather measure functional CB1 signaling at vHIP-BLA synapses using optogenetics, since changes in mRNA may not always translate to functional differences in CB1 signaling. Using this approach, we showed that CB1 agonist-induced synaptic depression at vHIP-BLA synapses did not differ between susceptible and resilient mice (Supplementary Fig. 12).

(6) Fig. 2a: Scheme is not clear. Days 6-9 are missing. The presentation implies a temporal progression, but it rather depicts there different blocks in the experimental set-up. The abbreviation FS-NC is not explained. Please improve this scheme.

Timeline has been clarified to include all days, and all abbreviations are explained within figure legends.

(7) Figure 8a-c: The authors should present the relevant comparison between Veh Res versus Veh Susc. Are there significant differences?

There are no differences in this comparison, and this has been added to the text.

(8) Figure 8, Suppl Figure 8: Although oDSE in vHIP-BLA projection is clearly altered in resilient versus susceptible mice (Figure 8), and although PreL-BLA oDSE is less strong than vHIP-BLA oDSE, it cannot be excluded that PreL-BLA oDSE is also different between resilient versus susceptible mice. Thus, this study did not formally prove that the 2-AG-CB1 signaling of the vHIP-BLA projection is responsible for the behavioral difference between resilient and susceptible mice. Only an association of altered oDSE with susceptibility was shown.

We have now added data demonstrating that PreL-BLA oDSE does not differ between resilient and susceptible mice (Figure 8). We also show that the strength of synaptic connectivity in this pathway is not different between susceptible and resilient mice and that tonic 2-AG signaling induced by JZL-184 is not different between susceptible and resilient mice. These data add specificity for the role of the vHIP-BLA circuit in modulating the observed phenotypes.

(9) Are there differences in DSE in BLA without optic stimulation between resilient and susceptible mice?

Importantly, the addition of our data examining PreL-BLA oDSE demonstrates a degree of specificity of this effect to the vHIP inputs. Given this specificity we did not compare electrical DSE as it is very likely that the effect of electrical stimulation (all inputs) could be masked by differential effects found at specific inputs (PL vs. vHIP), making the overall difference uninterpretable when using electrical stimulation approaches.

(10) The authors must mention and discuss some work on stress/traumatic stress - amygdala function - cannabinoids, performed by the group of Irit Akirav. This work fits to the present one, where THC alleviated the stress effects.
Done.

(11) The authors may wish to see this work in the context of other investigations on stress resilience and alterations in synaptic transmission / excitability, such as done by Bo Li and Eric Nestler.

We have mentioned Dr. Nestler's work on chronic social defeat stress and have now added Dr. Li's work on learned helplessness in the discussion section.

Minor points:

(1) Please term the behavioral outcome as "suppression of feeding" and do not term this behavior from time to time as anxiety behavior.

In the results section we have used the term "feeding latency" as this is the actual measurement we are taking. In the discussion it is our interpretation, supported by literature, the increases in feeding latency reflect increases in "anxiety-like behavior", and this have used this terminology in our interpretation of the meaning of changes in feeding latency.

(2) Page 23: Please specify type of LC-MS instrument used. Unfortunately, the authors did not use the current state-of-the-art method for endocannabinoid / lipid quantifications, which is MS/MS, whereby the identity of the ions can be clearly uncovered.

The title of the section has been changed to clarify the use of LC-MS/MS which was previously only indicated by the inclusion of the transitions used for selective reaction monitoring. We have also added the type of instrument.

(3) Figure 1, and all other figures: Please explain abbreviations always in the legend. In Figure 1: NIH.

Abbreviations have now been explained in each figure legend.

(4) Figure 1a-c: Explicitly mention the group size, as the individual points are not visible, as in particular the blue bars covers the blue dots.

Sample sizes have been increased and numbers indicating the group sizes have been added.

(5) Figure 1d,e, Figure 4a,b,e,f: The treatment with rimonabant is not very meaningful, as it is completely dominant in the influence on feeding latency and food consumption. Please comment. In Fig. 1, RIM was given at a dose of 1 mg/kg. Was the same dose given in the experiments shown in Figure 4?

The same 1mg/kg dose of Rimonabant was used in all experiments. This is quite a low dose and does not significantly impact NIH latency in the absence of stress. Moreover, we have previously shown that the effects of Rimonabant on latency and food consumption are dissociable; it decreases food intake under all conditions, but only increases latency under novel cage conditions (Gamble-George et al 2013). If the effects on latency were a consequence of decreased appetite, Rim should increase latency under all conditions also, but it does not. Thus we concluded that cannabinoids are able to regulate both latency (anxiety-like measure) and consumption (appetitive measure) independently. We feel these data are important to include as they speak to the receptor mechanisms by which JZL-184 exerts its behavioral effects.

(6) Figures 3, 4 etc.: Please indicate in the graphs the meaning of the x-axis (days). Also indicate by arrows the time points of shock delivery.

"Days" added to x-axes. Arrows added to indicate foot-shock.

(7) Figure 4d: Please add "Feeding" to the y-axis.

Done.

(8) Figure 4i: While for the resilient group the color is kept in red, for the logic for the black, blue and orange colors are not clear.

Within figures blue =JZL-184, orange=Rimonabant, and purple=DO34. Resilience is now exclusively indicated by black regardless of treatment condition.

(9) Figure 6c: The immunostaining for DAGL-alpha is of insufficient quality. This protein should show a clear postsynaptic staining, which is not visible in the "GFP" group.

This IHC was performed using an antibody Ken Mackie's lab has previously validated (see Katona et al. 2006 J. Neuro), and these images were only meant to show that the viral cre delivery was indeed effective at reducing DAGL expression. The main validation comes from the reduction in 2-AG levels measured in punches from the viral injection site also shown in Fig. 6. Moreover, detailed anatomy of DAGL α expression in the BLA has been extensively demonstrated in the literature (see work of Watanabe, Katona and others) and is not the focus of the

present work. Alternate images have been added to the figure to better show the degree of deletion of DAGL α as requested by reviewer 1.

Reviewers' comments:

Reviewer #1 (Remarks to the Author):

The authors present a well and thoughtfully revised manuscript, addressing the issues raised in the initial submission. With the exception of the suggested region-specific interventions, all of my previous comments were addressed satisfactory, including the addition of experiments and data demonstrating that the observed effects are independent of appetite and food intake. While it is a pity that the region-specific manipulation experiments using DAGLa overexpression have not worked out, I still believe that overall the manuscript presents compelling and complete evidence for the involvement of the endocannabinoid system in stress resilience and adaptation.

Reviewer #2 (Remarks to the Author):

I am satisfied with the authors responses, and have no further concerns. I believe these novel findings will substantially add to the literature.

Reviewer #3 (Remarks to the Author):

The authors addressed in the revision most of the critical points, in particular several control experiments.

However, to my surprise, and also considering the communication with the authors through the editor prior to the revision process, the authors did not address major point 4.

Instead, the authors operated 80 mice in order to overexpress DAGLalpha through AAV-mediated gene transfer into the amygdala, aiming to enhance 2-AG signaling. In principle, this would have been an alternative approach as compared to local injection of JZL184. However, this experiment did not work out and gave inconclusive results, possibly due to low transgene expression. In fact, it appears that viral overexpression is very challenging. The authors stated (again) that local application of JZL184 is outside their "current expertise". I still cannot follow this reasoning, as this technique is commonly used by many, many researchers, and does not require much more experience and expertise as compared to local AAV injections. Furthermore, the lack of time cannot be a reason, as in general, the editors allow prolongations in revision processes, in particular, if animal models are involved.

Clearly, this work presents a huge amount of highly interesting results. However, as it stands, the systemic pharmacological treatment with JZL184 cannot be linked to 2-AG signaling in the vHIP-BLA projection, or at least in the BLA, when it acts as a presumable resilience promoting factor. Exactly the local JZL184 application is the missing link and would have made this work brilliant. In the current version, rather strong statements can be made regarding the pathways and brain regions where dysfunctional 2-AG signaling leads to the susceptibility to traumatic stress. Thus, from the mechanistic point of view, the data presented strongly indicate that the lack of 2-AG signaling in the BLA makes the mice more susceptible, possibly/likely by a dysregulation of 2-AG signaling (as investigated by oDSE) in

the vHIP-BLA projection. On the other side, the general increase in 2-AG signaling (by systemic application of JZL184) makes the mice more resilient, but neither the responsible brain region(s) nor the pathways were identified in this work. Based on the fact that the lack of 2-AG in BLA makes mice more susceptible, it is not possible to make the conclusion (without experimental evidence) that enhancing this pathway in the BLA induces/enhances resilience. Therefore, the title of the paper promotes a lot of expectations, and is therefore misleading.

The second concern is regarding the behavioral paradigm. As exposed in major point 2, the magnitude of the stress effects decays rather fast as compared to other "resilience" paradigm, such as social defeat. This might be a matter to be critically discussed.

Reviewer #1 (Remarks to the Author):

The authors present a well and thoughtfully revised manuscript, addressing the issues raised in the initial submission. With the exception of the suggested region-specific interventions, all of my previous comments were addressed satisfactorily, including the addition of experiments and data demonstrating that the observed effects are independent of appetite and food intake. While it is a pity that the region-specific manipulation experiments using DAGLa overexpression have not worked out, I still believe that overall the manuscript presents compelling and complete evidence for the involvement of the endocannabinoid system in stress resilience and adaptation.

We understand the limitations regarding the lack of drug microinjections, and have added a specific section in the discussion describing the limitations of the work in that regard (page 23).

Reviewer #2 (Remarks to the Author):

I am satisfied with the authors' responses, and have no further concerns. I believe these novel findings will substantially add to the literature.

We thank the reviewers for their efforts in helping us to improve the quality of our manuscript.

Reviewer #3 (Remarks to the Author):

The authors addressed in the revision most of the critical points, in particular several control experiments. However, to my surprise, and also considering the communication with the authors through the editor prior to the revision process, the authors did not address major point 4. Instead, the authors operated 80 mice in order to overexpress DAGLalpha through AAV-mediated gene transfer into the amygdala, aiming to enhance 2-AG signaling. In principle, this would have been an alternative approach as compared to local injection of JZL184. However, this experiment did not work out and gave inconclusive results, possibly due to low transgene expression. In fact, it appears that viral overexpression is very challenging. The authors stated (again) that local application of JZL184 is outside their "current expertise". I still cannot follow this reasoning, as this technique is commonly used by many, many researchers, and does not require much more experience and expertise as compared to local AAV injections. Furthermore, the lack of time cannot be a reason, as in general, the editors allow prolongations in revision processes, in particular, if animal models are involved.

Clearly, this work presents a huge amount of highly interesting results. However, as it stands, the systemic pharmacological treatment with JZL184 cannot be linked to 2-AG signaling in the vHIP-BLA projection, or at least in the BLA, when it acts as a presumable resilience promoting factor. Exactly the local JZL184 application is the missing link and would have made this work brilliant. In the current version, rather strong statements can be made regarding the pathways and brain regions where dysfunctional 2-AG signaling leads to the susceptibility to traumatic stress. Thus, from the mechanistic point of view, the data presented strongly indicate that the lack of 2-AG signaling in the BLA makes the mice more susceptible, possibly/likely by a dysregulation of 2-AG signaling (as investigated by oDSE) in the vHIP-BLA projection. On the other side, the general increase in 2-AG signaling (by systemic application of JZL184) makes the mice more resilient, but neither the responsible brain region(s) nor the pathways were identified in this work. Based on the fact that the lack of 2-AG in BLA makes mice more susceptible, it is not possible to make the conclusion (without experimental evidence) that enhancing this pathway in the BLA induces/enhances resilience. Therefore, the title of the paper promotes a lot of expectations, and is therefore misleading.

The second concern is regarding the behavioral paradigm. As exposed in major point 2, the magnitude of the stress effects decays rather fast as compared to other “resilience” paradigm, such as social defeat. This might be a matter to be critically discussed.

While we appreciate the reviewer’s perspective regarding local drug microinjection studies, we feel the work contains strong evidence for the conclusions as stated, and does provide evidence that BLA DAGL α is required to express physiological resilience to repeated stress exposure. In collaboration with the journal editors, we decided to add a specific section in the discussion stating that a limitation of the study is that the site of action of systemic 2-AG augmentation is not known and may involve the amygdala or other brain regions as well, and that further studies using region-specific DAGL α overexpression or pharmacological approaches would be needed to clarify this issue (page 23). We have additionally changed the manuscript’s title to be certain it is not misleading. The new text in the discussion section is highlighted in blue. We have also added specifications in the abstract and results that pharmacological treatments were systemic in nature.

We agree that it is important to be clear about the decay of the phenotype and we have added a sentence about this to the discussion (page 20).